# Hydrogeological conceptual model of andesitic watersheds revealed by high-resolution heliborne geophysics

Benoit Vittecoq[1,2], Pierre-Alexandre Reninger[3], Frédéric Lacquement[3], Guillaume Martelet[3], Sophie Violette[2,4]

[1]BRGM, 97200 Fort de France, Martinique
[2]ENS-PSL Research University & CNRS, UMR.8538 – Laboratoire de Géologie, 24 rue Lhomond, 75231 Paris France
[3]BRGM, F-45060 Orléans, France
[4]Sorbonne Université, UFR.918, F75005, Paris France

*Correspondence to*: Benoit Vittecoq (b.vittecoq@brgm.fr)

**Abstract.** We conducted a multidisciplinary study at the watershed scale of an andesitic-type volcanic island in order to better characterize the hydrogeological functioning of aquifers and to better evaluate groundwater resource. A heliborne TDEM survey was conducted over Martinique Island in order to investigate underground volcanic structures and lithology, characterized by high lateral and vertical geological variability, and resulting in a very high heterogeneity of their hydrogeological characteristics. Correlations were made on three adjacent watersheds between resistivity data along flight lines and geological and hydrogeological data from 51 boreholes and 24 springs, showing that the younger the formations, the higher their resistivity. Correlation between resistivity, geology and transmissivity data of three aquifers is attested: within the interval 10-100 ohm m and within a range of 1 to 5.5 Ma the older the formation, the lower its resistivity, and the older the formation, the higher its transmissivity. Moreover, we demonstrate that the main geological structures lead to preferential flow circulations and that hydrogeological watershed can differ from topographical watershed. The consequence is that, even if the topographical watershed is small, underground flows from an adjacent watershed can add significant amounts of water to such a catchment. This effect is amplified when lava domes and their roots are situated upstream, as they present very high hydraulic conductivity leading to deep preferential groundwater flow circulations. We also reveal, unlike basaltic-type volcanic islands, that hydraulic conductivity increases with age in this andesitic-type volcanic island. This trend is interpreted as the consequence of tectonic fracturing associated to earthquakes in this subduction zone, related to andesitic volcanic islands. Finally, our approach allows characterizing in detail the hydrogeological functioning and identifying the properties of the main aquifer and aquitard units, leading to the proposition of a hydrogeological conceptual model at the watershed scale. This working scale seems particularly suitable due to the complexity of edifices, with heterogeneous geological formations presenting high lateral and vertical variability. Moreover, our study offers new guidelines for accurate correlations between resistivity, geology and hydraulic conductivity for volcanic islands. Finally, our results will also help stakeholders toward a better management of water resource.

## 1 Introduction

Water resources management on volcanic islands is challenging as these territories are often densely populated, subject to several natural hazards (volcanism, earthquakes, tsunamis, landslides, erosion and sea level rise, etc.), and with increasing water demands (for irrigation, drinking water, etc.) or overexploitation of rivers or aquifers. Understanding the hydrogeological
functioning of these islands is thus a major issue to achieve a sustainable management of their water resources. Hydrogeology of volcanic islands is challenging taking into consideration the complexity of these edifices and the difficulties encountered when acquiring accurate in-situ data (such as steep slopes, tropical vegetation, few access tracks, distance from laboratories, extreme climatic and hydrometric conditions for equipment, etc.). Indeed, as exposed by Ingebritsen et al., (2006), volcanic formations exhibit extreme spatial variability or heterogeneity, both among geologic units and within particular units, with
large variation from core scale to regional scale, permeability being, especially in volcanic environment, a scale-dependent property.

Historically, basaltic islands have been widely studied: (e.g. Hawaii: Peterson, 1972; Macdonald et al., 1983; Canarian Islands: Ecker, 1976; Custodio et al., 1988; Custodio, 2005; Custodio and Cabrera, 2008; Cruz-Fuentes et al., 2014; Izquierdo, 2014; Iceland: Sigurðsson and Einarsson, 1988; Réunion Island: Violette et al., 1997; Join et al., 2005; Azores Islands: Cruz and
Silva, 2001; Cruz, 2003; Galapagos Islands: d'Ozouville et al., 2008; Pryet et al., 2012; Violette et al., 2014; Jeju Island: Hamm et al., 2005; Won et al., 2005, 2006; Hagedorn et al., 2011; or Mayotte: Vittecoq et al., 2014), leading to several hydrogeological conceptual models, essentially at the island scale, each model being intrinsically dependent on the dynamic of volcanism activity, on the number and history of volcanoes and their effusive and rest phases, generating a more or less complex geometry within which water infiltrates and circulates in a complex pattern, according to the recharge conditions.
Andesitic islands in subduction zones, and especially the Caribbean ones, are less known and a limited number of hydrogeological studies have been conducted and published in these archipelagos, mainly at the island scale (e.g. Unesco, 1986; Falkland and Custodio, 1991; Davies and Peart, 2003; Gourcy et al., 2009; Vittecoq et al., 2010; Robins, 2013; Hemmings et al., 2015). Charlier et al. (2011) showed the interest in working at the watershed scale to define a hydrogeological scheme of a tiny site (45 ha) in Guadeloupe Island. Hydrogeological analyses of volcanic formations at several scales are
indeed essential, especially for andesitic volcanism, characterized by heterogeneous geological formations, with alternation between intense eruptive phases marked by andesitic lava flows, pyroclastic flows, lahars, etc. interspersed with quieter phases marked by the dismantling of the volcano with debris avalanches and meteoric and alluvial erosion (Westercamp et al., 1989; 1990). Furthermore, andesitic stratovolcanoes display volcanic facies trends with variation and lateral distribution between central, proximal, medial and distal zones, depending on the valley and interfluve dynamics (Vessel and Davis, 1981; Bogie
and Mackenzie, 1998; Selles et al., 2015). Finally, meteorological and hydrothermal weathering processes are superimposed on these lithological heterogeneities. This high lateral and vertical geological variability thus induces a very high heterogeneity of their hydrogeological characteristics. As shown by most of these studies, without in-depth data, it is not possible to understand relevant geological structures and consequently to understand the hydrogeological functioning.

Recently, heliborne geophysical surveys (e.g. Sorensen and Auken, 2004) started providing new regional in-depth data, which contribute to solving this scientific and technical challenge. High-resolution heliborne EM (ElectroMagnetic) resistivity data provide information down to the first hundred meters along flight lines, and allow a continuous imagery of resistivity variations. Geological structures and hydrogeological properties can then be interpreted from these geophysical data to determine and constrain accurate conceptual models. To be relevant, and because resistivity is not a univocal parameter, this dataset analysis must be constrained with as much direct observation data (outcrop, borehole geological log, hydraulic conductivity data, etc.) as possible (see for instance Vittecoq et al., (2014) for Mayotte basaltic island).

Vittecoq et al., (2015), in studying an andesitic coastal aquifer in Martinique, demonstrate the relevance of working and analyzing heliborne EM data at the aquifer scale, to characterize geological and hydrogeological heterogeneities of a 15 Ma old geological formation. At this scale, this approach is corroborated thanks to a very long term pumping experiment. The working scale should indeed be sufficiently fine to be relevant to the structural specificities of these andesitic volcanic islands. However, working scale should also include surface and hydrogeological watersheds to integrate water balance estimation, interaction between groundwater and surface water, potential contribution of different aquifers and vertical downward transfers, for a comprehensive view of the water cycle, so that stakeholder can use the results for a sustainable management of water and energy resources.

Considering these issues, we conducted a multi-disciplinary approach at a watershed scale, based on the correlation of geological, hydrological, hydrogeological and heliborne Time Domain ElectroMagnetic (TDEM) data. We focus on a few strategic watersheds situated in Martinique, a predominantly andesitic volcanic island (Westercamp et al., 1989) located in the Lesser Antilles volcanic arc, in the subduction zone between the Atlantic plate and the Caribbean plate. The goals of our study are thus to: (i) characterize the structure and hydrogeological functioning of Martinique andesitic aquifers at the watershed scale, (ii) show the influence of geological structures on groundwater flows and the consequence on the interactions between rivers and aquifers, (iii) assess the adequacy and difference between hydrological watershed and hydrogeological watershed, (iv) propose a conceptual model at the watershed scale, and (v) strengthen the hypothesis of Vittecoq et al., (2015) that, in contrast with the basaltic islands, hydraulic conductivity may increase with age in andesitic-type volcanic islands.

## 2 Martinique Island and studied watersheds

### 2.1 Site location and climate

Martinique Island (Fig. 1) is the largest volcanic island (1 080 km²) of the Lesser Antilles Archipelago. Its relief is mountainous in the North (highest volcano at 1 397 m) and gentler in the South (highest hill at 504 m). Rainfall is characteristic of a humid tropical climate controlled by the trade winds and orographic effects (Guiscafre, 1976; Vittecoq et al., 2010), with the rainy season between July and November and the dry season between January and April, interspersed with fluctuating transition periods. Annual temperatures vary between 18°C and 32°C at Fort-de-France and an East trade wind regime ensures relatively

constant ventilation. Average annual precipitation (Fig. 1C) is high in the northern part, reaching 5 000 to 6 500 mm/yr at the summits, and between 1 200 and 1 500 mm/yr in the south.

The three studied watersheds (Fig. 1) are located just near the capital city of Fort-de-France whose urban area includes half the population of the island (376 500 inhabitants on the island in 2016). Three dams are located on the Case Navire River, and

provide an average of 5.9 $10^6$ m³/yr to the urban area. During the driest seasons, the river is often dry over several hundred meters downstream of the dams, causing strong environmental impacts. Consequently, scientific researches are expected to understand the hydrological and hydrogeological functioning of this area, in order to propose alternative water resources management.

## 2.2 Geology

The volcanic activity of Martinique Island (Westercamp et al., 1989; Germa et al., 2010; 2011), which began more than 25 Ma ago, is characterized by a succession of many volcanic formations, mainly andesitic, set up from a dozen principal volcanic edifices, active during successive phases, with alternating periods of construction and erosion, sometimes contemporary.

The geology of the study area (Fig. 2A and S1) is concerned with two distinct phases and volcanic edifices (Westercamp et al., 1989): the Morne Jacob shield volcano and the Carbets volcanic complex (Fig. 2B). The Morne Jacob Shield Volcano is

the largest edifice on the island and lasted 3.3 Ma. Given its position, offset from the pre-existing reliefs, the first phase is first submarine then aerial. First phase formations are mostly weathered, because of a long period of rest and erosion of at least 1 Ma before the next phase. The strong aerial effusive volcanic activity of the second phase of the Morne Jacob volcano is witnessed on the field by massive flows (²α) up to 200 m thick. The Carbets volcanic complex develops on the western flank of the Morne Jacob shield volcano and lasted 1.8 Ma with four main aerial phases.

Despite this detailed knowledge of the nature and location of the geological formations constituting the watersheds, and their lateral extension at the 1:50000 scale; it remains difficult to have a precise and 3D vision of their geometries and relationship at depth.

## 2.3 Hydrogeology

The position of the springs and the available drilling data (Fig. 1 and Fig. 2, Table S1 and Table S2) suggest that aquifers could
be associated with almost every volcanic phase of each edifice.

### 2.3.1 Cold springs

Springs (Fig. 1, 2 and Table S1) are located mainly in the upper part of the watersheds. Springs water discharge are most of the time a few liters per second. They are associated with four main geological formations (Fig. 3A and 3B). Seven springs, situated between 440 and 580 m amsl, emerge from andesitic and dacitic dome and lava flows ⁹αbi (0.3 to 0.9 Ma). This
geological formation is the last main event of the Pitons du Carbet Complex, strongly marking the landscape with several monolithic domes. In addition to observed springs, many perennial rivers flow from these peaks, so the aquifer that feeds these

springs and rivers can be considered as an important perched aquifer. Three springs, situated between 473 and 505 m amsl, emerge from andesitic lavas [8]ρα (0.9-1.2 Ma). Nine springs, emerging between 135 and 631 m amsl, are associated to andesitic lavas [2]α (2.2-2.8 Ma), and four springs, emerging between 296 and 350 m amsl, are associated with basaltic lavas [1]βol (4-5.5 Ma). These springs are mostly situated at slope foot, at slope breaks or at the top of gullies. Andesitic lavas [2]α, [8]ρα and basaltic lavas [1]βol are thus permeable and considered as aquifer formations. Finally, one spring emerges from debris flow ([6]B) associated to the first phase of construction of the old Carbet (2 Ma).

### 2.3.2 Thermal springs

Two thermal springs, Didier (210 m amsl – 32°C – 1 850 µS/cm) and Absalon (350 m amsl – 36°C – 1 730 µS/cm) are situated in the middle of the Case Navire watershed (Fig. 1 and 2), at a distance of 1700 m from each other. Their waters are mainly bicarbonated Ca-Na-Mg and are associated with high emissions of magmatic $CO_2$ and precipitation of iron hydroxides (Gadalia et al., 2014). The geochemical model (Gadalia et al., 2014) proposes an evolution in three stages: (1) deep mixing between water of meteoric origin and marine water (around 0.1%), during a first partial chemical and isotope equilibrium; (2) water-rock and magmatic $CO_2$ interaction at medium temperature (90-140°C) in a residual geothermal system, and (3) mixing with fresh waters during the ascent, at a lower temperature. The Absalon spring emerges within fissured and fractured andesitic lavas [2]α. The geological context of Didier spring is poorly known because the bottling plant masks the outcrops. A borehole drilled 200 m from the spring shows, under a thickness of 16 m of pyroclastic flow, andesitic lavas [2]α over 80 m thick. Waters of those two springs are thus mixed with the waters of the aquifer of andesitic lavas [2]α.

### 2.3.3 Boreholes

Fifty-one boreholes (Table S2) were drilled on theses watersheds or in the close vicinity (Fig. 1D and 2A). Transmissivity data are available for 19 boreholes ([1]βol, [1]α, [2]α, and [6]B) and vary by two orders of magnitude between $1\ 10^{-5}$ m² s⁻¹ and $1\ 10^{-3}$ m² s⁻¹, with an average value of $5\ 10^{-4}$ m² s⁻¹ (standard deviation: $3\ 10^{-4}$ m² s⁻¹). Hydraulic conductivity varies between $2\ 10^{-7}$ m s⁻¹ and $3\ 10^{-5}$ m s⁻¹ with an average value of $1\ 10^{-5}$ m s⁻¹ (standard deviation: $9\ 10^{-6}$ m s⁻¹). As aquifers are fissured or fractured with heterogeneities along the screen height, and as data were calculated by dividing transmissivity by the height of the screened saturated aquifer, calculated hydraulic conductivities have to be considered as minimum values.

Piezometric level measurements (Fig. 3A) show that the piezometric level is on average seven meters below ground level and attests that the hydrogeological functioning is not marked by a basal groundwater body with low hydraulic gradient. In addition, two main typology of aquifer are distinguished on Fig. 3B: on one hand perched aquifers with springs located in altitude above 400 m amsl and on the other hand aquifers crossed by boreholes in the valleys with water level close to the ground level.

Piezometric levels monitoring (Fig. 4) put in evidence unconfined aquifers (Piezometers 1, 2 and 4), with annual dynamics and well-defined seasonal cycle (with fluctuations between 1 and 2 m): low groundwater levels occur during dry seasons (April to July) and high ones during rainy seasons (August to December). In contrast, piezometer 3 (situated 1 kilometer above piezometer 2), characterizes a confined aquifer with multiannual dynamics, with a minor influence of seasonal cycle.

### 2.3.4 Water balance

The water budget hydrological terms of the studied watersheds have been computed in Fig. 5 in order to show a synthetized view of the annual water balance and the contribution of each hydrological term. Rainfall and potential evapotranspiration are provided by the national meteorological agency for the period 1991-2015 (the annual rainfall map is shown on Fig 1C). River discharge is monitored by the Ministry of Environment. Real evapotranspiration and effective rainfall are 1 km² spatialized data calculated by Arnaud and Lanini, (2014) (over the period 1991-2010), following a methodology detailed in Vittecoq et al., (2010) and based on the Thornthwaite model. The ratio runoff/infiltration and groundwater contribution to river discharge have been calculated (1) for Case Navire River by Vittecoq et al., (2007) (over the period 1987-1990) based on inverse modeling (e.g. Pinault and Schomburgk, 2006) with Tempo Software (Pinault, 2001) and (2) for Alma River by Stollsteiner and Taïlamé, 2017 (over the period 2010-2015) based on lumped hydrologic modelling (e.g. Thiery, 2010) using Gardenia software (Thiery, 2014). Both methods were using daily meteorological data series (rainfall, potential evapotranspiration) and rivers flowrates.

The Alma watershed is the highest and smallest one, located upland, and is exclusively covered with tropical forest. This watershed is equipped with a gauging station with valid data since July 2010 (specific discharge of about $112\,l\,s^{-1}\,km^{-2}$). Water balance calculation (Fig. 5) evidences that the difference between total effective rainfall and average annual flow in the Alma River is about $2.3\ 10^6\ m^3/yr$ (18% of effective rainfall volume). This volume of water (1) infiltrates in depth and / or (2) joins another stream / nearby hydrological watershed, if the hydrogeological catchment area differs from the topographic catchment. This volume infiltrated in depth or flowing towards an adjacent catchment area is therefore to be considered as a minimum value, as measured rainfall gauges are situated at elevations not exceeding 600 m whereas the watershed peak culminates at 1 197 m. The national climatic agency (Météo-France) considers that values up to 7 000 mm/yr of rain are quite possible values on the summits. Considering this highest value, and the various uncertainties on the water budget parameters, the deep infiltrated volume could reach a maximum of $8\ 10^6\ m^3/yr$.

The Fond Lahaye watershed culminates at 532 m of elevation and its stream joins the sea 4 km downstream. Since there is no gauging station on this river, it is difficult to define a water balance. In the maximalist hypothesis where 100% of the effective rainfall returns to the river, its maximum specific discharge would be of about $11\,l\,s^{-1}\,km^{-2}$ (corresponding to 10% of the nearby Alma watershed specific discharge).

The Case-Navire watershed culminates at 1 197 m of elevation and its stream joins the sea 10 km downstream. Its upstream part is divided into two sub-basins (Duclos and Dumauzé rivers) that meet 5 km before reaching the sea (Fig. 3B). Three dams are located in the upstream part of the Case Navire River (one on the Dumauzé River and two on the Duclos River, cf. Fig. 1). The annual volume of the three dams on arrival at the main distribution tank is $5.9\ 10^6\ m^3/yr$ (over the period 2009-2012), corresponding to an average of 16 300 m³/day (and corresponding to 19% of the annual effective rainfall). During the driest seasons, the river is often dry downstream of the dams, causing strong environmental impacts. The gauging station is situated on the Case Navire River few hundred meters before reaching the sea (Fig. 1 and 2A), 5 to 6.5 km downstream the dams,

which allow calculating water balances (Fig. 5). The supposed natural flowrate of the Case Navire River is about 18.7 $10^6$ $m^3$/yr, corresponding to 60% of the annual effective rainfall, by adding water abstraction volume by dams. Consequently, the volume of groundwater circulating in this watershed that does not return to the river is about 12.4 $10^6$ $m^3$/yr. This volume may infiltrate in depth and circulates in the aquifers, to another watershed or flows into the sea.

These water balance calculations evidence the main key component of hydrological cycle of each watershed and provide first evaluations of groundwater budget. In particular, they reveal significant quantities of deep infiltrated water (14.7 to 20.4 $10^6$ $m^3$/yr), equal to two or three times the surface water intakes in the Case Navire River. There is thus a necessity to better understand aquifer nature and hydrodynamic characteristics, extension, thickness and groundwater preferential flows and interactions with rivers, and to locate recharge areas, in order to propose appropriate hydrogeological conceptual models,

necessary to a sustainable management of water resources.

### 3 – Heliborne TDEM method

Our methodology is based on a multidisciplinary approach combining geology, hydrogeology and a heliborne TDEM geophysical survey, in order to identify relationships between ground-based punctual geological and hydrogeological data on one hand, and in-depth geophysical information derived all over the area on the other.

**3.1 The survey**

A heliborne TDEM survey was conducted from February to March 2013 with the SkyTEM 304 system (Sørensen and Auken, 2004) over the entire Martinique Island. This survey, fully described by Deparis et al., (2014) and Vittecoq et al., (2015), was supervised by BRGM (the French geological survey) for geological and hydrogeological purposes. Over the studied watersheds, the survey was flown mainly along the N-S direction with 400 m line spacing, and along the W-E direction with

4 000 m line spacing. The spacing between each EM sounding along flight lines is approximately 30 m. In the lower part of the watershed less to no data have been acquired because of the urbanization. Finally, 13 596 TDEM soundings were processed in the study area. The TDEM method allows imaging the conductivity/resistivity contrasts of the subsurface, inducting eddy currents in the ground (Ward and Hohmann, 1988). Locally, the Depth Of Investigation (DOI) of the method depends on the emitted magnetic moment, the bandwidth used, the subsurface conductivity and the signal/noise ratio (Spies, 1989). In this

study, the average depth of investigation is around 150 to 200 m.

**3.2. TDEM data processing**

The ground clearance of the loop was obtained degrading an available 1 m Digital Elevation Model to a 25 m grid (consistently with the AEM footprint) and subtracting it to the DGPS altitude of the frame; we did not use the data from the laser, which proved to be noisy in such rough relief environment. Tilt measurements were processed taking into account the local

topography in order to consider an effective tilt at each TDEM data location (Reninger et al., 2015). As part of an environmental

study in an entropized area, particular attention was paid to properly remove noise from the TDEM data. They were processed with a singular value decomposition (SVD) filter (Reninger et al., 2011). The SVD allows explaining a dataset with only few components, each data being a linear combination of these components. Thanks to this decomposition we are able to identify and remove several types of noise, making the processing less time consuming and subjective and reducing the amount of

careful editing. In addition, a trapezoidal stack (Auken et al., 2009) was applied on the data. The trapezoid shape is consistent with the increase of the footprint of the EM method with time. The stack size was adapted to the noise level along flightlines. Thanks to this filter we manage to recover some noisy windows, which are unusable otherwise (Reninger et al., 2018). The aim of the applied processing was to keep as much resolution as possible (Reninger et al, 2018). Finally a manual editing was performed, mainly to remove remaining inductive/galvanic coupling noises. In order to improve the coverage of the dataset,

good quality portions of ferry lines were also considered during the processing (Reninger et al., 2018). Figures 1 and 2 display the position of the TDEM dataset after processing. Data were then inverted using the Spatially Constrained Inversion algorithm (SCI) (Viezzoli et al., 2008). Each TDEM data was interpreted as a 1-D earth model (EM sounding) divided into n layers, each one being defined by a thickness and a resistivity. During the inversion, constraints were applied vertically and horizontally on nearby soundings (independently of the flightlines and the ground clearance), weak constraints were applied for this study

in order to limit the smoothing of the inversion procedure. Results were obtained with a smooth inversion (consisting of 23 layers from 0 to 200 m depth). This inversion method is effective to image complex geological structures with the lowest dependency on the starting model. In addition, altitude of the transmitter was inverted for, and the DOI was evaluated, as a final step of the inversion (Christiansen and Auken, 2012).

## 4 – Resistivity profiles and correlations between resistivity, geological and hydrogeological data

Five resistivity profiles obtained inverting TDEM data are provided on Fig. 6 and 7 (localization in Fig. 2). Confronting these profiles with geological and hydrogeological data (springs, boreholes, observations and outcrops, geological map, etc.), the TDEM data can be interpreted in terms of geological or hydrogeological contrasts, and evidence the main internal geological structures and associated aquifers, at depth up to around 200 m. Thus, as exposed by Vittecoq et al., (2014) and Vittecoq et al., (2015), geological and transmissivity data of each borehole can be compared to the closest TDEM sounding in order to get

information on the resistivity of the aquifers and aquitards and better constrain their extension and thickness. However, in such particularly rugged and contrasted environment, it must be paid attention how this comparison is achieved, mainly in terms of distance and elevation. This was done on 18 boreholes. They are located at an average distance of 35 m (with a maximum distance of 90 m) to the closest EM sounding, with a difference in elevation less than 10 m. At each of these TDEM soundings, we looked at the average of the resistivity falling in each associated borehole geological formations. To complete aquifer

characterization, a specific analysis was conducted on the springs. Resistivity values of the cells located upstream the 24 springs (Table S1), corresponding to supposed aquifer formations, were manually extracted from the 3D resistivity models.

Figure 8A displays borehole (BR) and spring (SR) aquifer resistivity ranges. Alluvial deposits display a relatively large resistivity range (12-74 ohm m) because of the heterogeneity of alluvial materials (in terms of granulometry, nature, ages…). Except for alluvial deposits, a good correlation appears between resistivity and the age of the geological formations, showing the relationship between weathering process and resistivity: the older the formation, the lower its resistivity. Correlation between geology and hydrodynamic properties (Fig. 8B) also displays a trend: the older the formation, the higher its transmissivity or its hydraulic conductivity. In particular, the relatively large resistivity and hydraulic conductivity range for andesitic lavas [2]α could be related to their intrinsic heterogeneity. These correlations are relevant for the three studied aquifers, within the interval 10-100 ohm m and within a range of 1 to 5.5 Ma. Below 10 ohm m, several authors (e.g. (d'Ozouville et al., 2008; Pryet et al., 2012; Vittecoq et al., 2014) put in evidence that very low resistivity layers can correspond to high permeability formations saturated with saltwater (old confined water or seawater intrusion) or to impermeable clays resulting from meteorological or hydrothermal weathering processes. Beyond 100 ohm m, there are no boreholes on the studied watershed, with transmissivity or hydraulic conductivity values, crossing formations with resistivity values higher than 100 ohm m. The correlation between age and hydraulic properties is valid for the same kind of rocks (ie. Andesite and basalt lava flows in the context of subduction zone volcanic arc island) but cannot be considered for domes. Indeed, eruptive mechanism of andesite and basalt flows on one hand, and intrusive domes on the other hand are different, and domes are only observed in this area between 0.3 and 0.9 Ma.

These correlations demonstrate the necessity and advantage of coupling hydrogeological data (springs, boreholes…), geological and geophysical data for an advanced interpretation of resistivity data, as such information is scarce in volcanic islands environments and because alone resistivity data does not allow differentiating age, nature of geological formations or aquifer identification.

## 5 – Hydrogeological conceptual model

Our methodology and associated correlations allows identifying and characterizing the main aquifer and aquitard formations (synthetized in table 1) as well as their lateral extent and thickness, enabling the construction of a hydrogeological conceptual model at the hydrogeological watershed scale. This conceptual model, synthetized on Fig. 9, characterizes the structure and hydrogeological functioning of andesitic aquifers at the watershed scale, and highlights the influence of geological structures on groundwater flows and the consequence on the interactions between rivers and aquifers. Joint analysis of water balance and geological structure also allows putting in evidence the differences between hydrological watershed and hydrogeological watershed.

### 5.1    The upper major perched aquifer of andesitic domes

The conceptual model is marked by the presence of andesitic domes and lava flows ([9]αbi), occupying the upper part corresponding to half of Case Navire watershed and the entire Alma watershed. Water balance calculated on Alma River

suggests that 85% of effective rainfall (Reff) infiltrates in these andesitic domes. Considering the high resistivity values of the domes (Cf. Fig. 8A, springs resistivity analysis: 150-300 ohm m) and in comparison with other volcanic islands (d'Ozouville et al., 2008; Pryet et al., 2012; Vittecoq et al., 2014), it is assumed that these andesitic domes ($^9\alpha bi^D$) are highly fissured and fractured, conferring a high hydraulic conductivity to this aquifer (with an order of magnitude of $7 \cdot 10^{-5}$ m s$^{-1}$ by similarity with a borehole drilled in a dacitic dome 6 km north from the studied watershed). Given the rooting of endogenous domes within the volcano, and as shown thanks to the water balance calculation (Fig. 5), up to 40% of Reff seeps in depth within domes roots, through fissures and fractures, and recharge underlying aquifers.

In addition, the unsaturated zone should present significant thickness, and since some springs have relatively low flow rates, we consider that they could emerge thanks to small and low hydraulic conductivity horizons, such as paleo-soils, or geological heterogeneities (for instance between $^9\alpha bi$ and underlying formations), or structural discontinuities. The main rivers have their sources in this important perched aquifer, with significant flow rate (as show in Fig. 5, for instance the Alma River specific discharge is 112 l s$^{-1}$ km$^{-2}$). On the western and eastern topographic ridges of the Case Navire River watershed, andesitic lava flows ($^9\alpha bi^C$) also constitute the first aquifer receiving rainfall and from which flow some non-perennial springs during the rainy season, and few perennial springs during the dry season.

## 5.2    The lower aquifer of andesitic lavas

In this conceptual model the upper major perched aquifer, described above, underlies the second main aquifer of thick andesitic lavas $^2\alpha$, marked by a relatively "smooth" morphology or paleo-topography of their top, consistent with the structure of lava cooling along a shield volcano. Hydraulic conductivity data dispersion over two orders of magnitude is in agreement with the heterogeneity of theses andesitic lavas. The various facies that were observed at the outcrop (S1) are: (1) auto-brechified breccias and lavas, (2) massive facies more or less fractured according to the cooling rate of the lava, (3) facies with flow structures showing significant horizontal cracking parallel to the substratum, and (4) breccias and scorias associated with the base of the lava flow. Tectonic fracturing superimposes on these heterogeneities and can contribute to maintain and develop the hydraulic conductivity of volcanic formations, as shown by Vittecoq et al., (2015). In this type of andesitic formation, boreholes can also be dry, if no fissured or permeable zone is intersected.

The recharge of this aquifer is quite atypical as in the upper part of Case Navire and Dumauzé watersheds, effective rainfall is high, and permeable andesitic domes and lava flows $^9\alpha bi$ overlay andesitic lavas $^2\alpha$. As suggested by numerous springs in the limit of extension of $^9\alpha bi$, and the low flowrates observed in the two Fond Baron boreholes screened into andesitic lavas $^2\alpha$ (Senergues, 2014), effective rainfall infiltration into $^2\alpha$ should be limited by paleo-soils and/or the hydraulic conductivity contrast between the two formations, acting as semi-permeable hydraulic obstacles. In the lower part of the watersheds, effective rainfall is limited (200 to 800 mm/yr) compared to the upper part, and furthermore the plateau located on both sides of the rivers are overlain by low hydraulic conductivity breccias. Effective rainfall infiltration towards andesitic lavas $^2\alpha$ is thus also small in the lower part of the watersheds. Then, the recharge of this aquifer should follow four main steps. Firstly, a part of effective rainfall (18% to 40%, depending the watershed, as shown in Fig. 5) deeply infiltrates through the fractures

and in the rooting of andesitic domes [9]αbi. Secondly, as andesitic lavas [2]α were crossed through faulting by [9]αbi lavas, this deeply infiltrated water then flows deeper towards andesitic lavas [2]α, thanks to geological heterogeneity inside the old volcanic chimney. Thirdly, groundwater flows into andesitic lavas [2]α and lastly, the [2]α aquifer, incised by the river, allows this deeply infiltrated water to be drained by the river and the sea.

## 5.3 The regional aquitard

Hyaloclastites [1]H, mainly observable on Fig. 6 (C3) and Fig. 7 (C4 and C5) at altitudes below 100 m amsl, are the lower boundary of the watersheds and more generally, of a major northern part of the island. On Fig. 6 (C2), they are suspected between 200 and 300 m amsl on the east of the cross-section, probably due to the displacement generated by major faults: this topographical limit is interpreted by Boudon et al., 2007 as the eastern limit of a large flank collapse with a horseshoe-shape structure opened westward. The weathering grade observed on the outcrop in the Case Navire River, associated to their very low resistivity, lead to consider the hyaloclastites mainly as a very low permeable formation and are then considered as the regional aquitard.

## 5.4 Difference between hydrological watershed and hydrogeological watershed

The continuity of andesitic lava flows [2]α along the resistivity cross-sections (Fig. 7), from north to south and especially under the "Morne Jeanette" (C5), clearly suggests a continuity of groundwater flows, through andesitic lava permeable facies, beyond the Duclos River watershed and in direction of the Fond Lahaye watershed. This hypothesis of a clear difference between hydrological watershed and hydrogeological watershed is supported by (1) the piezometric fluctuations (Fig. 4), showing that Fond Lahaye upper borehole is in a captive aquifer with multiannual dynamic fluctuations, (2) groundwater mineralization and long duration time transfers (> 50 yr by CFC groundwater datation, Gourcy et al., 2009) and (3) the high flowrates of Fond Lahaye and Case Navire Boreholes (more than $1.2 \times 10^6$ m$^3$/yr have been calculated by Ollagnier et al., 2007; Vittecoq et al., 2008; Vittecoq et Arnaud, 2014).

## 5.6 Geothermal insights

The very low resistivity (6-10 ohm m) of hyaloclastites [1]H, cannot correspond to actual salt water intrusion, as they are situated higher above sea level. Their very low resistivity could rather result from weathering during the 1 Ma period of rest before the next volcanic phase and from hydrothermal weathering. . This low resistivity layer (< 10 ohm m) should indeed be an evidence of a smectite bearing hydrothermal altered caprock (e.g. Browne, 1970, Simmons and Browne, 1990) of an underneath geothermal system. The two thermal springs (Didier – 32°C, 1 850 µS cm$^{-1}$ and Absalon – 36°C, 1 730 µS cm$^{-1}$) could be leaks of this geothermal system, through faults allowing the rise of mineralized gaseous waters (it must be noticed that the supposed fault interpreted in Fig. 6 (C2 – 2 200 m) is aligned with Didier springs, Absalon springs and the Alma and Dumauzé Dacitic Domes).

Then, geothermal fluid circulations could follow five steps: (1) deep infiltration of effective rainfall through andesitic domes ($^9\alpha$bi) and associated deep rooting, (2) deep mixing at temperature between 100 and 140°C, according to Gadalia et al., (2014), (3) interaction with $CO_2$ and ascent along faults, (4) mixing with andesitic aquifer $^2\alpha$, and (5) emergence in thermal springs. The flowrate of these springs being relatively low, we can state that a part of the ascending enriched fluids do not emerge at the surface, and diffuse in the andesitic aquifer $^2\alpha$. The higher groundwater mineralization downstream (1 000 µS cm$^{-1}$ in Fond Lahaye boreholes), compared to the range of water electrical conductivity of cold springs and rivers (50-200 µS cm$^{-1}$) emerging from the aquifers upstream (cf. Table 1), clearly support this hypothesis.

## 6 – Discussion

Heliborne TDEM data reveals in depth resistivity contrasts. Their interpretation with boreholes and springs data allowed constraining a detailed hydrogeological conceptual model. Working at the watershed scale brings new elements of hydrogeological functioning of andesitic volcanic complex. Vessel and Davis (1981), Bogie and Mackenzie (1998) and Selles et al., (2015) proposed a geological conceptual model of andesitic stratovolcanoes putting in evidence central (0-2 km from the vent), proximal (5-10 km), medial (10-15 km) and distal (15-40 km) facies variations. The originality of our work is to focus on improving hydrogeological functioning of central and proximal parts of such andesitic system. Indeed, medial and distal parts, on which hydrogeological researches are generally focused on continental volcano (Selles, 2014), corresponding to lower and accessible area, are in our case under the sea.

On the scale of the island of Martinique, the proposed hydrogeological functioning conceptual model (and also our methodology) could likely been extended to the other watersheds situated on the Carbet volcanic complex and on the Morne Jacob shield volcano. Extrapolation to the entire Martinique is nevertheless not considered, as a specific hydrogeological functioning has been demonstrated for the centre of the Island (Vittecoq et al., 2015), as effective rainfall is significantly lower (<1500 mm) in the central-southern half of the island (Vittecoq et al., 2010) and because our conceptual model, concerning mainly fissured and fractured lava, could not fit with the Mount Pelée stratovolcano located in the North of Martinique (covering 15% of the Martinique area) and constituted by pyroclastic flows (Traineau et al., 1989).

Secondly, our conceptual model could also enhance, with new insights, existing characterization of the hydrogeology of small volcanic Islands and especially West Indies and Caribbean volcanic Islands (Unesco, 1986; Falkland and Custodio, 1991; Davies and Peart, 2003, and Robins, 2013). However, Hemmings et al., (2015), studying Montserrat Island, an andesitic island located in the Lesser Antilles, concluded that they didn't know which model between Hawaiian and Canarian model could fit with Montserrat island. As both were concerning only basaltic islands, with different geological structure and recharge conditions, other models have to be proposed. Our conceptual model, thanks to the high-resolution heliborne geophysical survey and correlations with geological and hydrogeological data, could then help better understanding hydrogeological functioning of other Lesser Antilles andesitic islands. For instance, the threefold division of West Indies hydrogeological

classification by Robins et al., (1990) could be updated with a fourth category considering groundwater in permeable perched high-rise volcanic dome and in underneath fractured volcanic rocks.

The main geological structures highlighted lead to preferential flow circulations and to a non-adequacy between hydrogeological and topographical watersheds, as supposed by Charlier et al., (2011) at a smaller scale (45 ha) in Guadeloupe.

The consequence is that even if the topographical watershed is small, underground flow circulations can add significant amount of water to river watershed's water balance, if aquifers are situated above (in elevation or upstream). We thus support the necessity to include and characterize neighboring watersheds to extend our methodology and results to others areas or islands. This can be even emphasized if lava domes and associated roots are situated upstream, as they present very high hydraulic conductivity and vertical in depth preferential flow circulations.

Thanks to the interpretation of the geological, geophysical and hydrogeological data, we highlight, for the present study (i.e. the watersheds and the three studied aquifers, within the interval 10-100 ohm.m and within a range of 1 to 5.5 Ma) that (1) the older the formation, the lower its resistivity and (2) the older the formation, the higher its transmissivity or hydraulic conductivity. This last result is also consistent considering the results of Vittecoq et al., (2015) obtained on an older aquifer (15 Ma) on Martinique Island, with higher hydraulic conductivity and lower resistivity than the ones observed in the present study. Consequently, unlike hot spot basaltic islands (Custodio, 2005; Vittecoq et al., 2014), hydraulic conductivity of the studied aquifers of subduction zone andesitic volcanism does not decrease with age. On the contrary, our results show an increase with age. Nevertheless, time itself is not the activating factor and only few geological processes can cause an enhancement of hydraulic conductivity. Given (1) the tectonic and seismic context of the subduction zone, (2) the fact that earthquakes are known for increasing hydraulic conductivity (e.g. Rojstaczer et al., 1995, Ingebritsen et al., 2006) and (3) the fact that earthquake induced modification of hydraulic conductivity have been observed in Martinique (Lachassagne et al., 2011), we interpret the observed hydraulic conductivity increase as the consequence of earthquake tectonic fracturing.

The accuracy of correlations between boreholes and TDEM soundings is highly dependent on the distance to the nearest TDEM flight line. Accordingly, a particular attention must be paid to the way this comparison is achieved, mainly in terms of distance and elevation difference. This being said, heliborne geophysical survey is certainly the best-cost efficiency method, and probably the only method providing this density of data down to 200 m depth, allowing a detailed geological and hydrogeological characterization at this working scale. Nevertheless a minimum of ground based geological and hydrogeological data are necessary, thanks to boreholes data.

**7 – Conclusion**

From an operational point of view, our data and results should be very helpful for local stakeholders facing environmental impacts and overexploitation of the Case Navire River. We show that large volumes of water infiltrate and flow in several aquifers. Sustainable management of water resources will require a better repartition between rivers and aquifers. Aquifers, and especially downstream the watersheds, could be exploited in order to decrease the use of the dams, especially in dry

seasons. Future drillings programs could be launched, considering our conceptual model. We also provide some insights about potential geothermal resources such as the pathway of deep infiltrated water through the roots of the andesitic dome, the presence of a low resistivity regional aquitard and the link with the thermal springs.

In conclusion, our multidisciplinary approach and results allow characterizing in detail the hydrogeological functioning and characteristics of the main aquifer and aquitard units, leading to the proposition of a hydrogeological conceptual model of an andesitic island at the watershed scale, putting in evidence the key role of geological structures and volcanic domes on groundwater flows. We also demonstrate, for the studied geological formations, that hydraulic conductivity increase with age in this andesitic-type volcanic island. Moreover, the working scale seems particularly suitable due to the complexity of edifices,

with heterogeneous geological formations presenting high lateral and vertical variability. Andesitic-type volcanic island being little known and studied, our work offers, in addition to the proposed conceptual model and thanks to the high-resolution heliborne geophysical survey, new guidelines for accurate correlations between resistivity, geology and hydraulic conductivity for other volcanic islands.

## 8 – Acknowledgments

This paper is a contribution of "DemosTHEM" and "Karibo" BRGM Research programs. Previous investigations made by BRGM over the studied watersheds were co-funded by CACEM, ODE and BRGM. The heliborne geophysical survey, called "MartEM" program (Deparis et al., 2014), was co-funded by BRGM, the FEDER funds for Martinique, the Regional Office for Environment Planning and Housing (DEAL), the Regional Council and the Water Office of Martinique (ODE). The authors would like to thanks the handling editor Gerrit H. de Rooij and the two reviewers (T. Izquierdo and an anonymous reviewer)

for their useful remarks and comments that improved the quality of our paper.

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

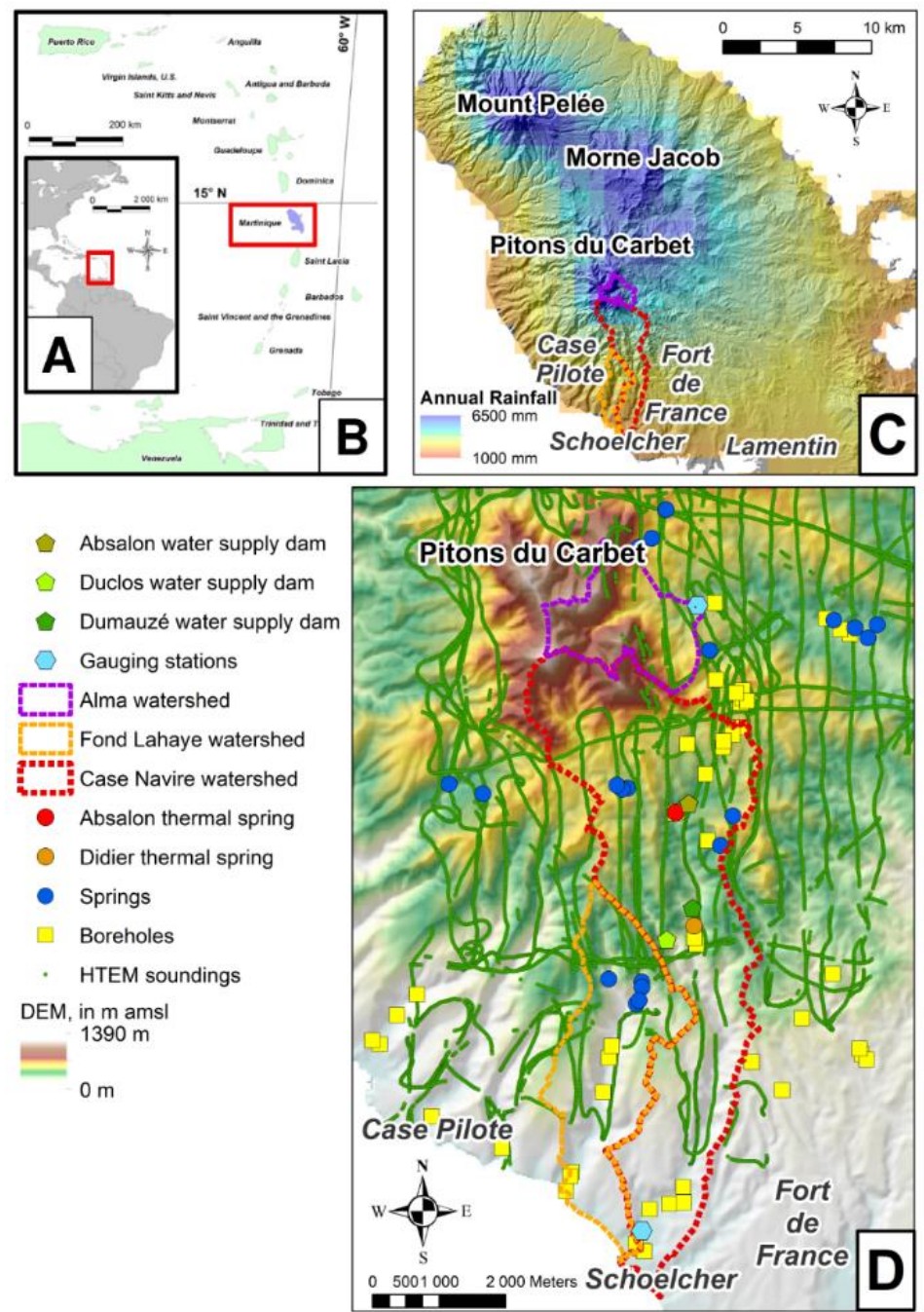

**Figure 1: The Location of the island of Martinique (A) on the scale of the America and (B) on the scale of the Lesser Antilles. Location of the watersheds (C) on the scale of the northern part of Martinique Island with annual rainfall map. (D) Location of rivers, water supply dams, gauging stations, watersheds, thermal springs, cold springs, boreholes and HTEM soundings along flight lines.**

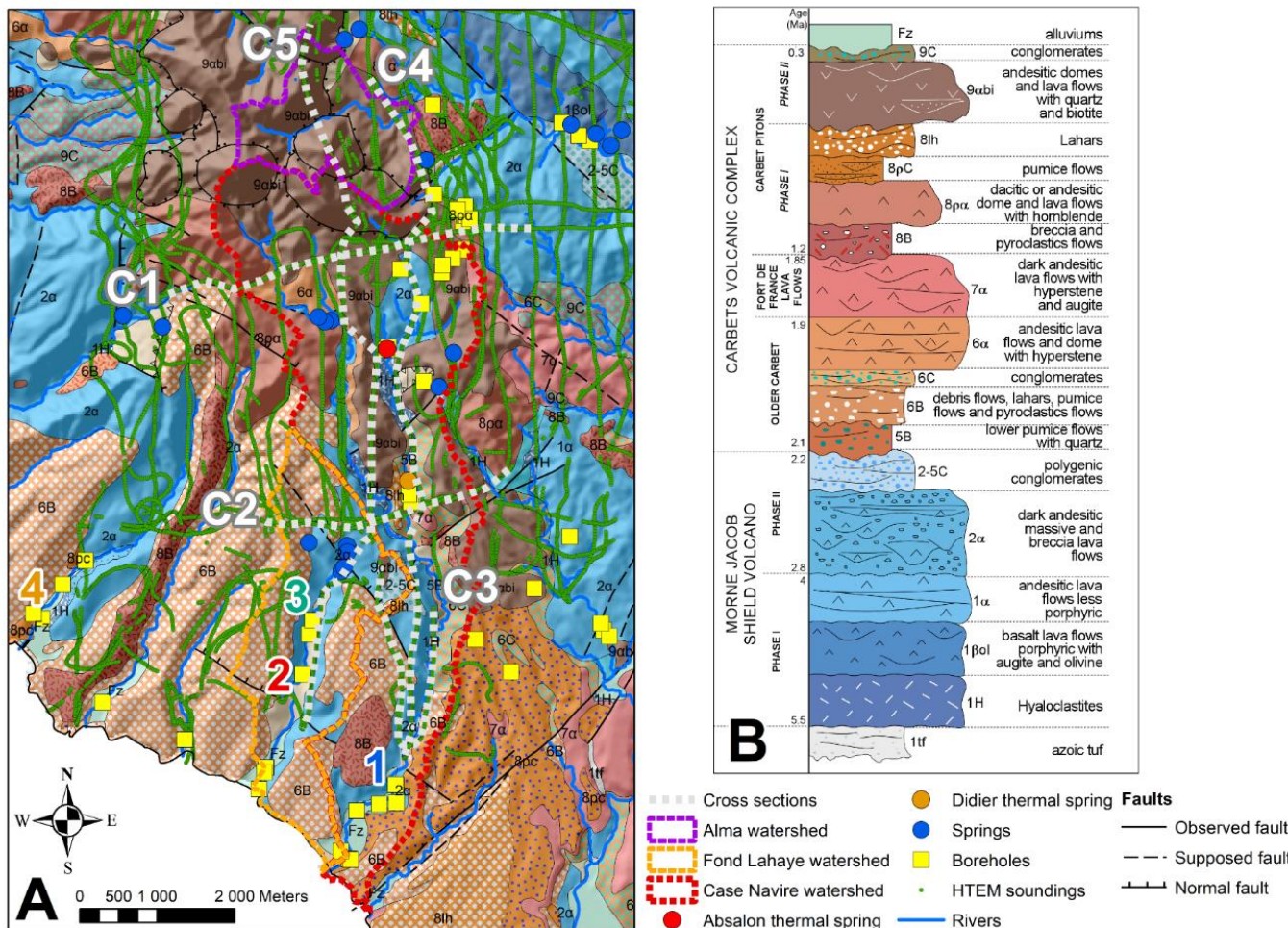

**Figure 2: (A) Geological map (adapted from Westercamp et al., 1990) of the studied watershedsLocation of the piezometers (piezometric chronicles on Fig. 4): (1) Case Navire, National number 1177ZZ0165, (2) Fond Lahaye National number 1177ZZ0161, (3) Fond Lahaye National number 1177ZZ0177 and (4) Case Pilote National number 1177ZZ0173. Cross-section location in white (Cross-sections on Fig. 6 and 7). (B) Litho-stratigraphic scale (adapted from Westercamp et al., 1990 and Germa et al., 2011).**

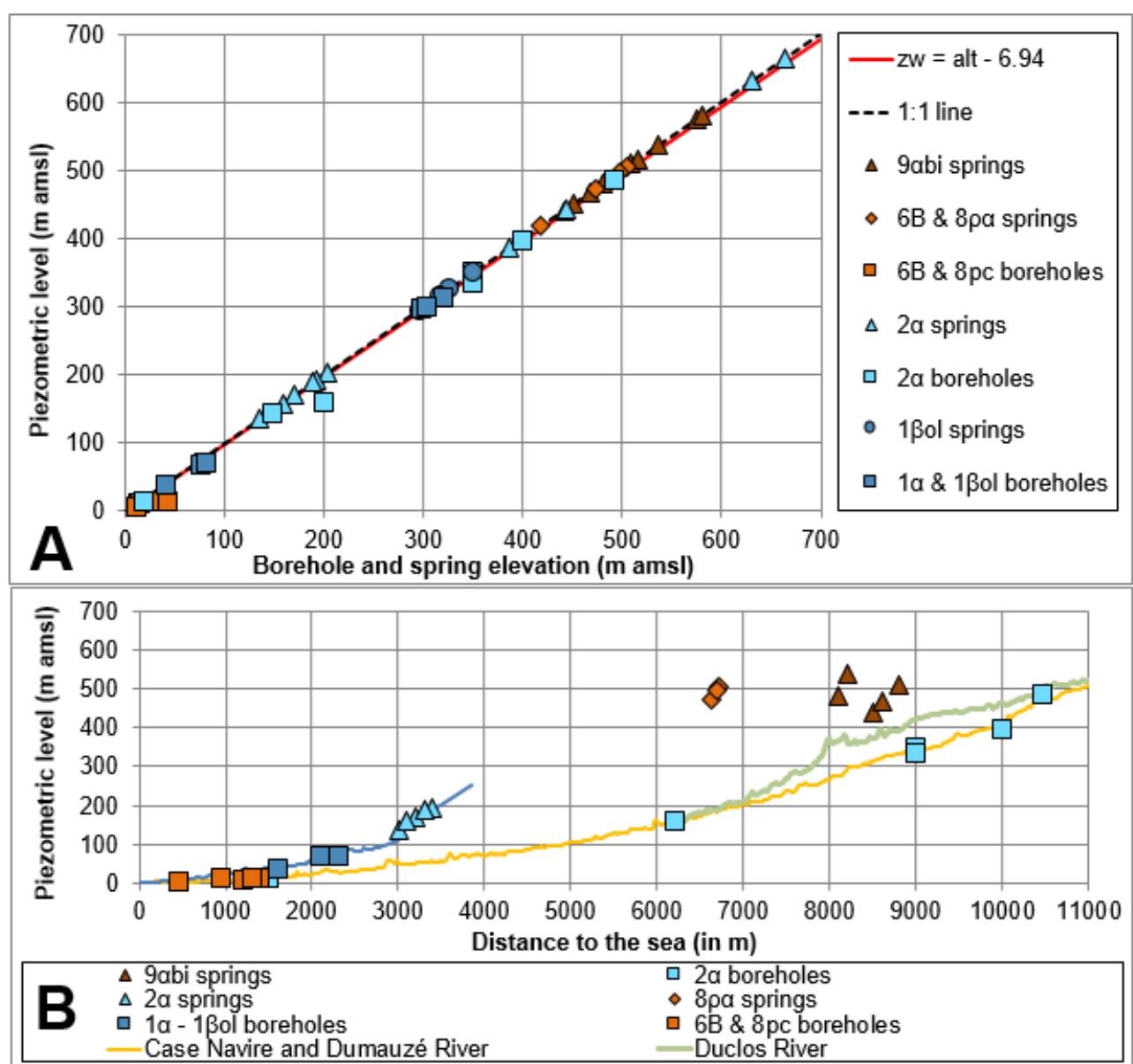

**Figure 3: (A)** Comparison between borehole and spring elevation and associated piezometric level (for twenty-six boreholes). The piezometric level is on average seven meters below ground level, following this linear relationship: zw = elev - 6.94 ($R^2$=0.99), where 'zw' is the piezometric level (m) and 'elev' the elevation (m). **(B)** Topographic profiles of Case Navire, Dumauzé, Duclos and Fond Lahaye rivers, the piezometric levels of boreholes in the associated watersheds with the lithology of the aquifer, and the elevation and lithology of the aquifer of springs.

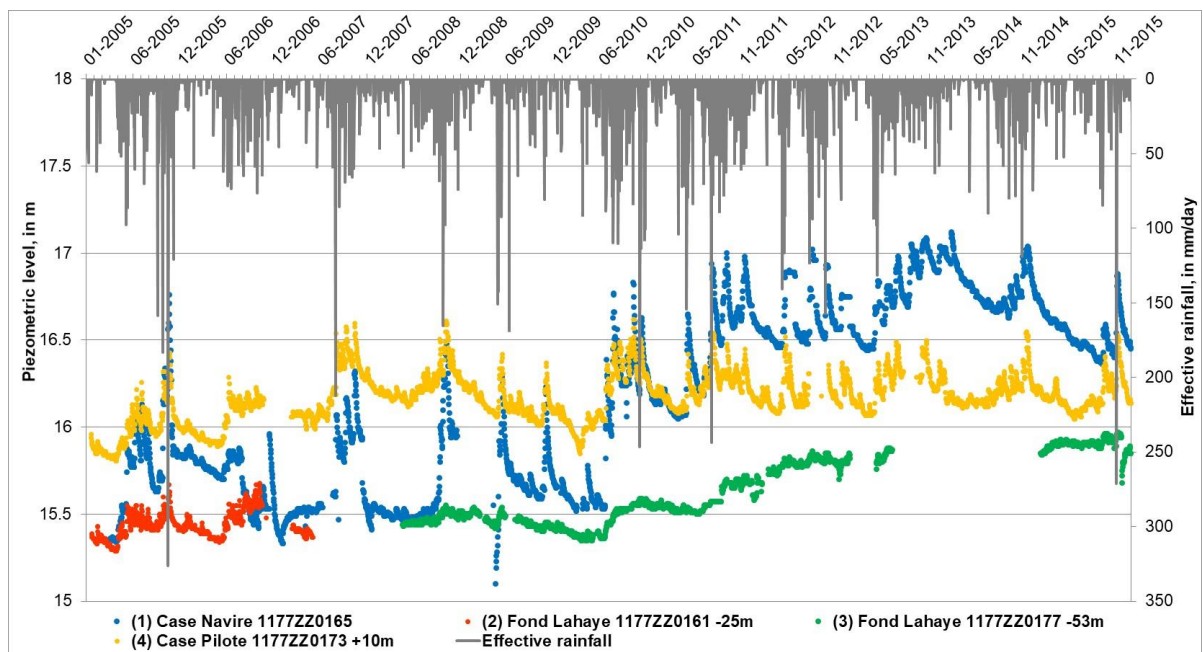

**Figure 4: Piezometric levels (monitored in the framework of the European piezometric network) and effective rainfall monitoring between 2005 and 2015. The first piezometer is located on the Case Navire watershed (1 on Fig. 2A), the two next are on the Fond Lahaye watershed (2 and 3 on Fig. 2A) and the last one is in Case Pilote city (4 on Fig. 2A), three kilometers west from Fond Lahaye. Piezometric levels of piezometers 2, 3 and 4 have been modified to fit on the same graph (-25m, -53m and +10m compare to their initial value, respectively).**

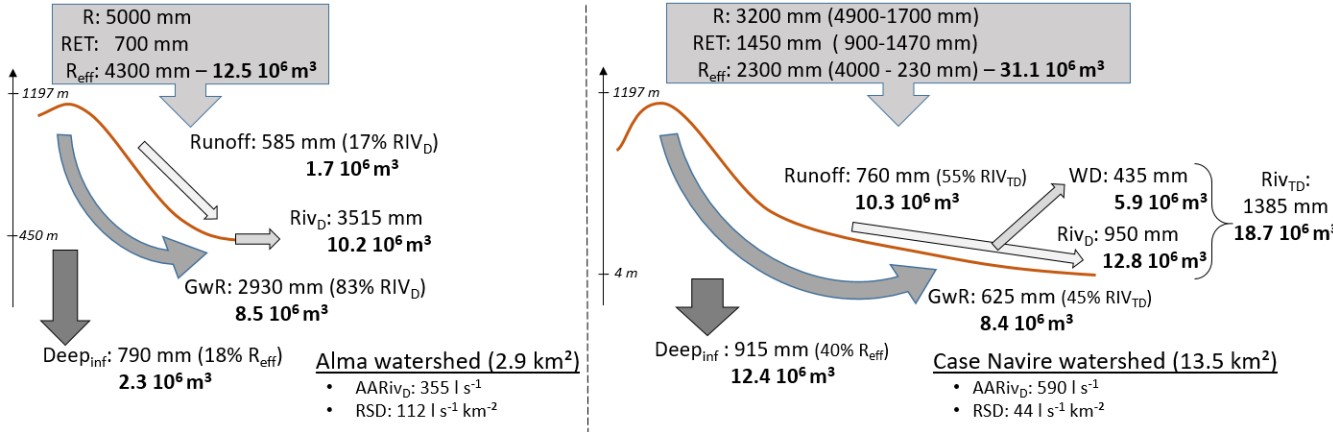

**Figure 5: Annual water balance of Case-Navire and Alma watersheds. Rainfall (R), Real Evapo-Transpiration (RET) and Effective Rainfall (Reff) are from Vittecoq et al., (2010) and Arnaud and Lanini, (2014). Average annual river discharge (AARivD) and river specific discharge (RSD) are calculated from gauging stations. Runoff: Reff contribution to river discharge. WD: volume of water for the drinking water distribution system. RivD: river discharge. RivTD: river total discharge (=RivD+WD). GwR: Groundwater contribution to river discharge. Deepinf: deep groundwater flow.**

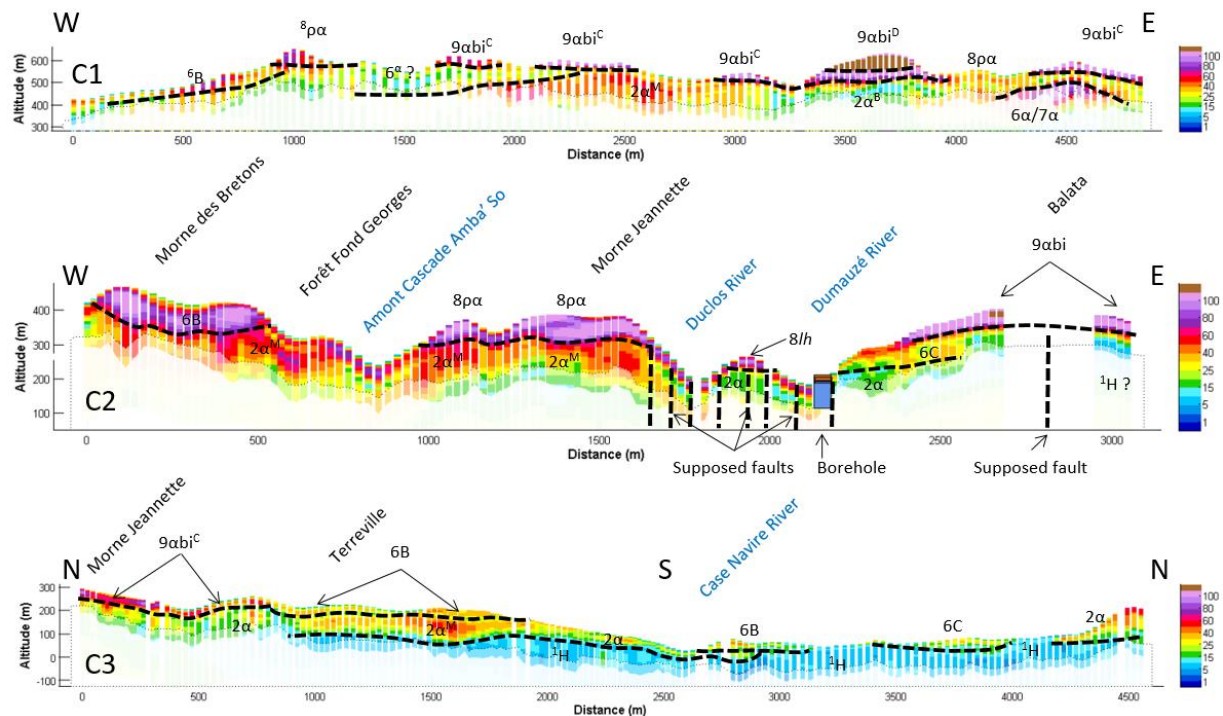

**Figure 6: Internal resistivity and hydrogeological structure along 3 cross-sections: C1, C2 and C3**

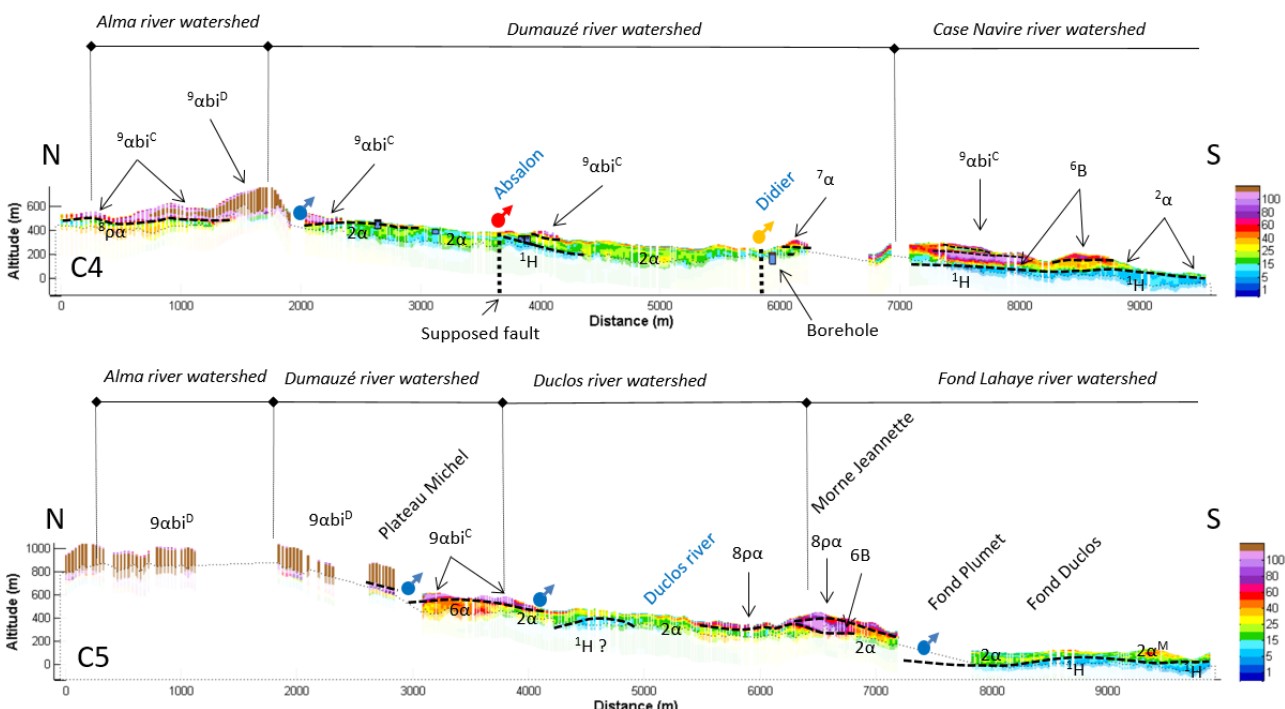

**Figure 7: Internal resistivity and hydrogeological structure along 2 cross-sections: C4 and C5**

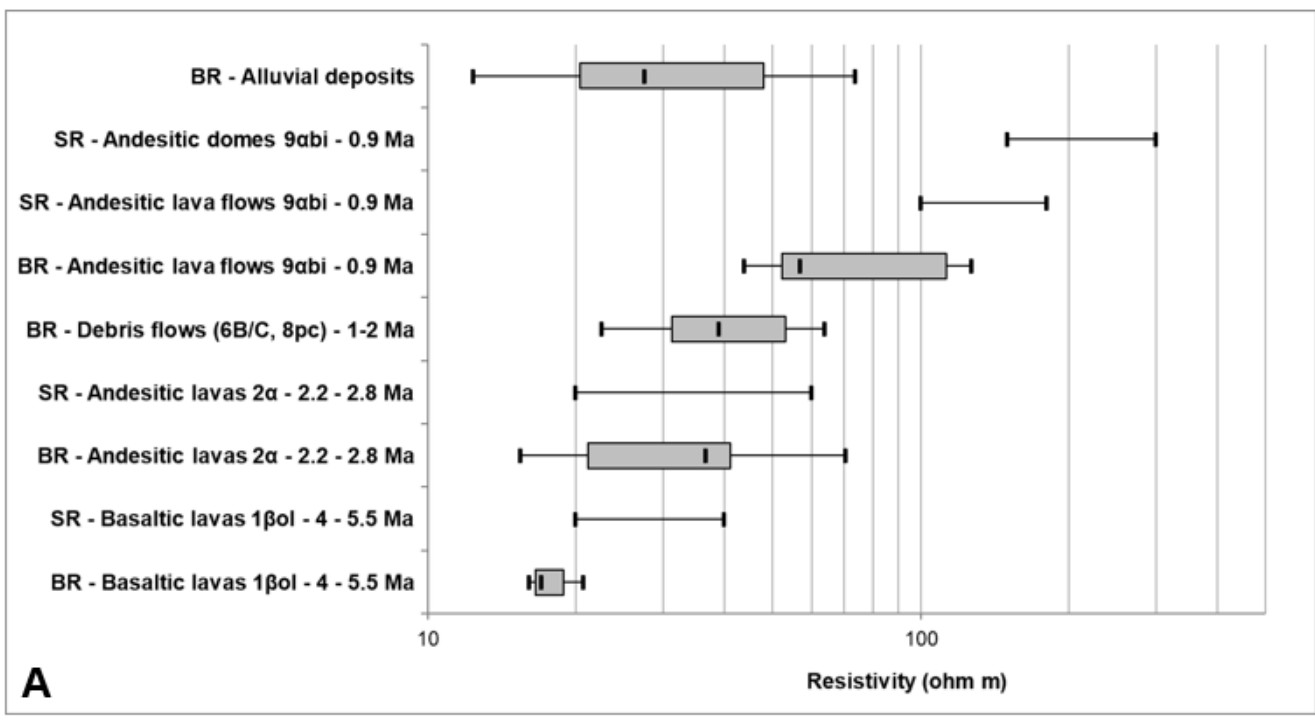

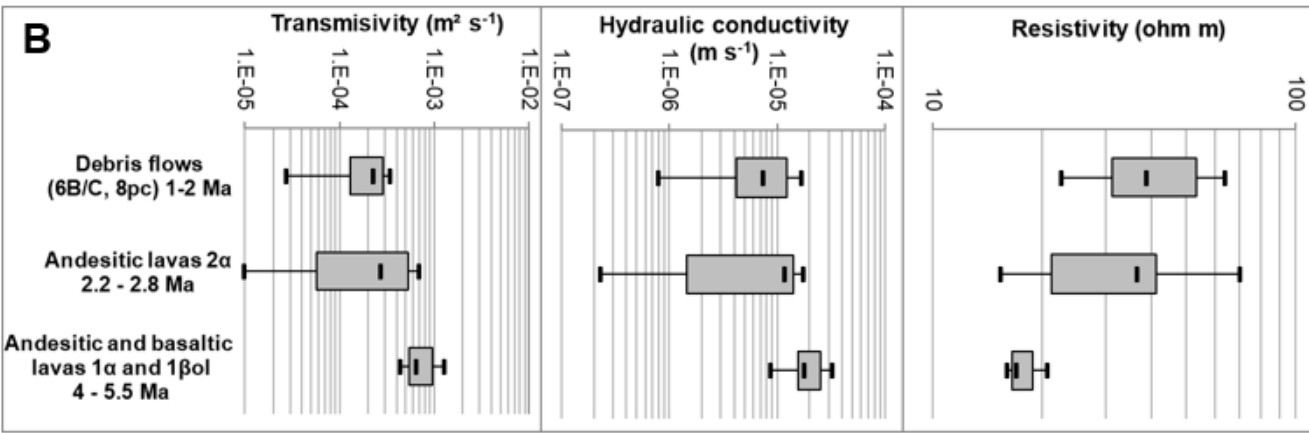

**Figure 8: (A)** Boreholes (BR) and springs (SR) resistivity ranges according to their lithological facies and age (Fig. 2). The younger the formation, the higher its resistivity. **(B)** Comparison between transmissivity, hydraulic conductivity and resistivity for three aquifer formations considering boreholes values.

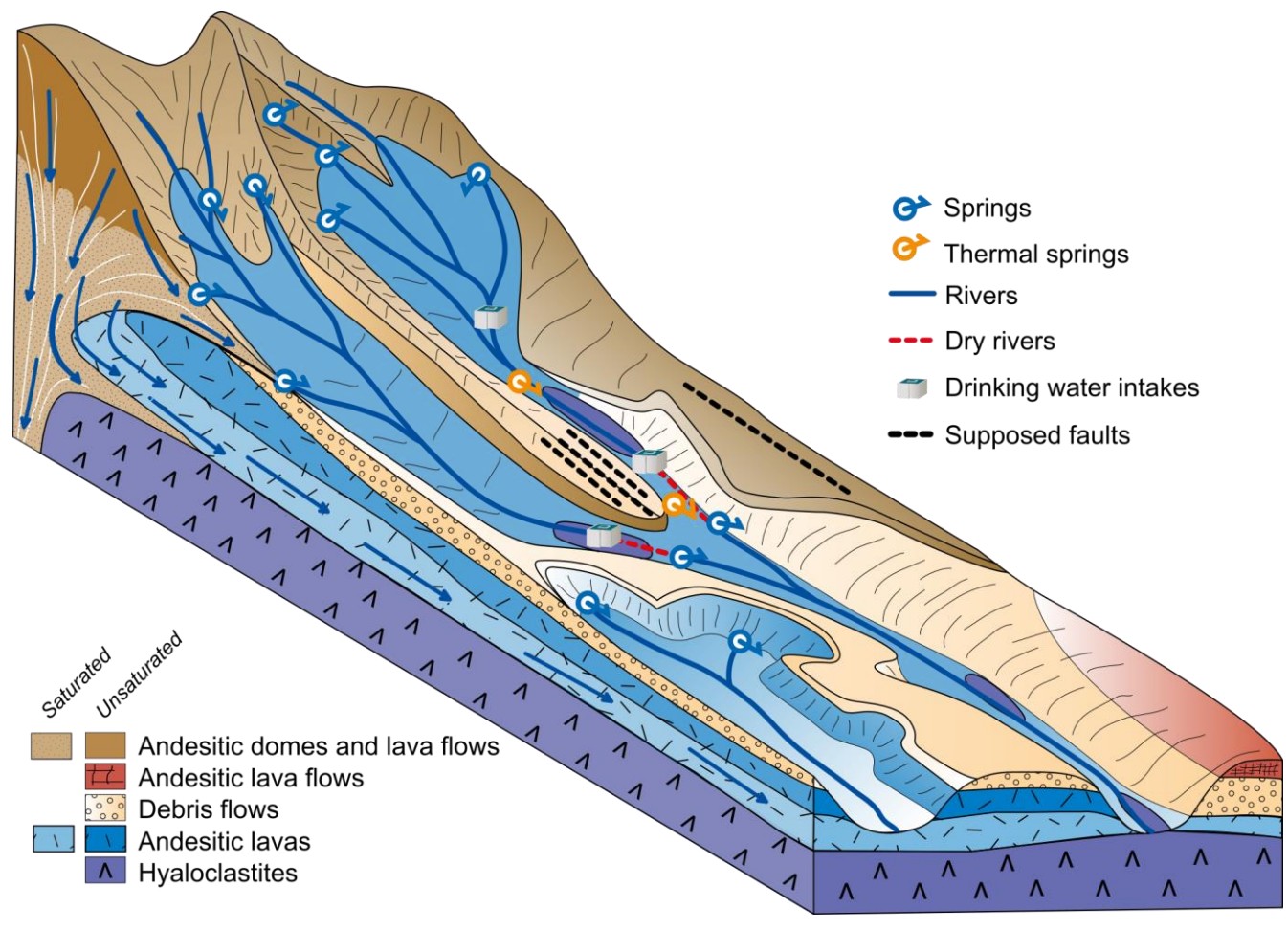

**Springs**

**Thermal springs**

**Rivers**

**Dry rivers**

**Drinking water intakes**

**Supposed faults**

Saturated

Unsaturated

Andesitic domes and lava flows
Andesitic lava flows
Debris flows
Andesitic lavas
Hyaloclastites

**Figure 9: Hydrogeological conceptual model of an andesitic complex in subduction zone at watershed scale.**

| Code | Lithological | Age (Ma) | Thickness (m) | Resistivity ranges ohm.m (Q1-Q3) | Aquifer typology | Porosity | Transmissivity (m$^2$ s$^{-1}$) | | | Hydraulic conductivity (m s$^{-1}$) | | Water electrical conductivity (µS cm$^{-1}$) | |
|---|---|---|---|---|---|---|---|---|---|---|---|---|---|
| | | | | | | | Nbr. of boreholes | range | Average and std. | range | Average and std. | Nbr. of springs | range |
| $^9\alpha bi$ | $^9\alpha bi^D$ - Andesitic domes | 0.3 - 0.9 | > 500 | Springs : 150 - 300 | Upper major perched aquifer | highly fissured/fractured | - | - | - | - | - | 7 | 100 - 160 |
| | $^9\alpha bi^C$ - andesitic lava flows | 0.3 - 0.9 | 10 - 100 | Springs: 100 - 180 Boreholes: 50 - 100 | Upper minor perched aquifer | highly fissured/fractured | - | - | - | - | - | - | - |
| $^8\rho\alpha$ | Andesitic and dacitic domes and lava flows | 0.6 - 1 | 10 - 100 | 70 - 120 | Minor perched aquifer | Fissured/fractured | - | - | - | - | - | 3 | 160 - 350 |
| $^6$B/C | Breccias, debris flow from the first phase of construction of the old Carbet | 2 | 100 - 200 | 20 - 40 | Minor perched aquifer | Heterogeneous Locally aquitard (cemented breccias) | 3 | 2.3 10$^{-4}$ to 3.7 10$^{-4}$ | 2.9 10$^{-4}$ std: 5.8 10$^{-5}$ | 6.7 10$^{-6}$ to 1.7 10$^{-5}$ | 1.1 10$^{-5}$ std: 4.6 10$^{-6}$ | 1 | 290 |
| $^2\alpha$ | Andesitic lava flows | 2.2 - 2.8 | 100 - 300 | - 2$\alpha^M$ (massive parts): 30 - 70 - 2$\alpha^f$ (fissured and fractured parts): 15 - 30 - 2$\alpha^b$ (brecchias and autoclastics parts): 10 - 15 | Major aquifer | Heteregeneous | 9 | 1.0 10$^{-5}$ to 7.0 10$^{-4}$ | 3.3 10$^{-4}$ std: 2.4 10$^{-4}$ | 2.4 10$^{-7}$ to 1.8 10$^{-5}$ | 9.0 10$^{-6}$ std: 6.4 10$^{-6}$ | 9 | 50 - 280 |
| $^1\alpha$ and $^1\beta$ol | Andesitic and basaltic lavas | 4 - 5.5 | 100 - 300 | 16 - 20 | Major aquifer | Fissured/fractured | 7 | 4.4 10$^{-4}$ to 1.3 10$^{-3}$ | 7.7 10$^{-4}$ std: 2.9 10$^{-4}$ | 8.8 10$^{-6}$ to 3.3 10$^{-5}$ | 2.0 10$^{-5}$ std: 7.5 10$^{-6}$ | 4 | 80 - 130 |
| $^1$H | hyaloclastites | 4 - 5.5 | > 200 | 6 - 10 | Regional aquitard | very low permeable formation | - | - | - | - | - | - | - |

**Table 1: Geological, geophysical and hydrogeological characteristics of the main aquifer and aquitard formations.**