# Peer review of "Hydrogeological conceptual model of andesitic watersheds revealed by high-resolution heliborne geophysics"

_Hydrology and Earth System Sciences, 2018_

## Short Comment (SC1) · 15 Feb 2019

This paper is really interesting, and it was a nice journey for me to read it in details, notably also as I worked for a long time in this island, and with one of the co-authors of the paper. This paper thus comprises a real potential for publication in HESS, particularly as it may present original results/theses on at least 4 different topics: 1. as it addresses the structure and functioning of an andesite-type volcanism (subduction-type) watershed, this kind of geological context being much less studied and documented than basaltic-type volcanism (hot spots, medio-oceanic ridges and transform faults, etc.); 2. as it comprises not only a classical geological approach to characterize the

structure of the aquifers, but also high-resolution airborne geophysics which is also not so common, particularly in andesitic volcanism; 3. as it tries to compute water budgets at the scale of surface/groundwater watersheds, which is not often performed in such a rather complex hydrogeological context and also notably as it requires data sets long and difficult to acquire; 4. and also and finally as it proposes an original thesis: andesite-type volcanic rocks hydraulic conductivity is assumed as increasing with the age of the rocks, contrary to some case studies of basaltic-type volcanic rocks. This process is interpreted by the authors as a consequence of tectonic fracturing that would develop a "cumulative" permeability, the rocks being not only fractured, but pervious, proportionally to the duration of their exposure to earthquakes, so to their age.

However, some inconsistencies in the data and/or in the interpretations are undermining the current proposed results. It is then necessary, to my opinion, that the authors improve the processing of their data, their interpretations and their argumentation, complete their data with additional ones if possible, to better support all or some of these hypotheses/theses, and thus to strengthen the stronger results of the paper before publication.

POINT N°2 As regards point N°2, the combined geological and airborne geophysical approach: The 1/50'000 scale geological map of Martinique, used and cited by the authors (Westercamp et al., 1989, 1990), is well known since now about 30 years to be a very high quality and highly reliable document. This is particularly the case in this Morne Jacob (MJ) shield volcano - Pîtons du Carbet (PdC) area where the later formations (PdC) cut and fill the earlier ones (MJ), and form and fill well identified paleovalleys; these paleovalleys roughly appear on Fig. 2a (orange/brown PdC formations lying on blue MJ formations). I am a bit disappointed as I was expecting that the combination of the new airborne geophysical data, and this high-quality preexisting map would help to produce much higher quality results, valuable for applied geology, and particularly hydrogeology, such as a higher quality geological map, precise geological cross sections and, why not, the mapping of the depth of the main geological

formations. Then, I feel that the geometry of the different geological formations could have been much better described. Moreover, deepening the interpretation of this data set would surely lead the authors to shift from some vague assertions about "heterogeneity", "preferential flow circulation", etc. to the precise geological identification of the structure governing groundwater flows, namely permeable structures on one hand, and impervious ones on the other hand. Globally, and as regards the hydrogeological conceptual modelling of volcanic aquifers (cf. Point N°1), it would help the authors to shift from rather old low resolution conceptual hydrogeological models (highlighted as such by the authors at lines 13, 19, 28-29, etc. of the manuscript) to the nowadays needed high-resolution hydrogeological conceptual modeling (see for instance Lachassagne et al., 2014 where this issue of low to high-resolution conceptual hydrogeological modelling in volcanic aquifers is also discussed in details). I would then suggest the authors to complete their current interpretation with an iterative approach: First: valorize in depth the existing geological map, and particularly draw precise cross-sections or, better draw 3D surfaces, to well delineate the main geological structures (paleovalleys, paleosurfaces, main lithological bodies below or within these paleovalleys, etc.). In that frame, the outputs from the quite recent paper from Germa et al. (2011) may surely be valorized more in depth.

Second: complete this geological model with the airborne geophysics and with the 51 boreholes cited in the text, most of them being not available at the time the geological map was published. For instance, one can see on profiles C1, C2 that a same geological formation (2alpha for instance) is not homogeneous, and comprises structures that can surely be interpreted: paleomorphologies, weathered levels, unweathered higher resistivity areas, etc.. It should also enable to discuss the resistivity range of these geological formations which is finally not so large (less than 5 to about 100 ohm.m). This process should also enable to calibrate much more in details the airborne geophysics with these geological and geophysical data. Then, it would be nice, as soon as this stage of the process, to highlight the outputs of the new airborne geophysics. These data may also be compared/completed with the available existing field geophysical

measurements (the first author of the paper participated to or directed several geophysical ERT campaigns on that area of Martinique that may be worth to be integrated in this paper.

Third and finally, I am sure that this systematic process will enable real progress in the knowledge of (i) the geological/hydrogeological structure of this part of Martinique such as: - identification of geological structures such as dipping, paleovalleys, various homogeneous lithological bodies, etc. within each preidentified geological formation, notably within the MJ hyaloclastites, the MJ shield volcano superimposed series of lava flows, and the PdC paleovalleys infilling, - identification and characterization (thickness) of the main paleosurfaces (particularly, if possible, in the MJ formations) and the associated weathering profiles, and (ii) the hydrodynamical properties of these formations.

It may surely lead to improve or even revise the proposed ranges of transmissivity and permeability, and at least to affect them to a larger set of rock types (see my other comment on that topic below).

POINT N°3 As regards point N°3, computation of water budgets at the scale of surface/groundwater watersheds, I have 2 main concerns. The first one deals with the uncertainty of all the measurements and computations required to obtain a surface/groundwater watershed budget: uncertainty on rainfall, particularly with the steep orographic gradients prevailing in this part of Martinique, on real evapotranspiration computation, then on effective rainfall, also on gauging data, etc.. To be convincing, the authors must appropriately compute error propagation, and not only provide numbers in their annual water balances (see for instance Fig. 5) but also uncertainty ranges around these numbers. The authors must also explain more in detail how the groundwater balance computations were performed (maybe there is some information to be extracted from contrasted hydrological years: average, dry, wet, etc.). But also how these data, spanning over several years were averaged, if some exceptional years, or events such as hurricanes and tropical storms were removed or not, etc.. In this area with steep pluvio-climato-hydrological gradients, how did they use the results and

discretized computations from Vittecoq et al., 2010? How did they use the one from Arnaud et al. (2013)? Additionally, it is not clear how "runoff" and "GwR" were computed, and if this sharing of river discharge provides added value to the paper. If not, it could be removed. Additionally, the river discharge curves should be provided, and discussed at least briefly.

More generally, the paper lacks from a "methods" section (if necessary with adapted references) that would enable to explain to the reader most methodologies used. Moreover, the use of different units to describe the same data (mm/y, m3/y, L/s) adds an unnecessary complexity. Additionally, "perspectives" should be discussed in the discussion section.

I tried to get more in details in the computations and have the following remarks. Alma watershed computations: - how the number of 8 Mm3/y (line 4, p. 6) is computed is hard to follow: it seems that it is only by replacing R = 5000 mm by another hypothesis R = 7000 mm. Again, this large difference in R suggests large uncertainties on the water budget. Maybe it would be more convincing to present, for each surface watershed, not only uncertainties as I proposed above, but also lowest and highest estimates of the different components of the water budget, and then to demonstrate that, in all cases (in most cases?), some water could be "missing", and could then recharge aquifers; - I don't understand the about 10% difference between "RivD"= 3515 mm = 323 L/s, and AARivD = 355 L/s.

Fond Lahaye watershed computations: - Line 7, p. 6: "its average specific discharge".

Case Navire watershed computations: - again, the computations are very hard to follow: 3200 − 1450 = 1750, not 2300 - for Case Navire RivTD (592 L/s) = AARivD (590 L/s). It wasn't the case for the Alma watershed

The second concern deals with the fact that the authors didn't consider some important known hydrogeological outflows issuing from neighbor areas from the studied area. Particularly, the "Attila" spring is flowing out from the PdC domes (9alpha_bi andesites

- 9ïĄạbi) less than 1 km west from the upper part of the Case Navire watershed. The discharge of this spring is about 1 million m3/year (30 L/s) (Lachassagne et al. 2003) which is far from negligible. This well-known spring from Martinique should at least be cited and integrated in the interpretations. In fact, this is, to my opinion, an important outflow from this aquifer, considered by the authors "as an important perched aquifer" (line 27). The authors should surely: 1. consider the boundaries of this lithological unit in their study, and then its inflows (recharge) and outflows, and also its relationships with the neighbor aquifers and surface watersheds; 2. hierarchize the springs, in order to distinguish high discharge springs, that constitute the real outflows from this aquifer ("regional outflows"), and low discharge springs that may be very local, and even that may be only representative of superficial formations in this context of very high rainfall. Additionally, presenting and discussing the geological context of the main springs (why they outflow where they outflow) may be very helpful. From this hierarchization of springs, the authors may revise their conceptual model (Fig. 9), and for instance not "plot" on it the very local springs. The same could be done for all springs (see again section 2.3.1.) emerging from the other geological formations. As far I remember, and as it is acknowledged by the authors in several parts of the paper, most of these springs have a rather low discharge. The co-ordinates of this Attila spring are the following: X = 701,550 (km); Y = 1627,240 (km) ; Z = 495 (m NGM)

Then, on Fig. 1 (or on a table to be provided that would support section 2.3.1.), the average discharge of each springs should be provided, as some springs, like Attila, are large discharge springs, but most others, as highlighted by the authors (line 23), are very local ones, with a very low discharge (a few L/s). In such a study that would like to get from the local to the watershed scale, not the same importance must be given to all springs.

Additionally to these concerns, to my opinion, an important and rather simple to acquire data set is missing in the paper. A survey of the streams during the dry season would surely have highlighted on one hand areas with outflows from the aquifers that feed

the rivers, on the other hand, areas with water leaks from the rivers to the aquifers; and also of course areas without interactions between aquifers (or lack of aquifer) and rivers. These inflows/outflows/no flow could (i) have served as real proofs of river water leaks, to strengthen the argumentation, and (ii) have been correlated to the lithology. It would also have fulfilled the need for a high-resolution approach in such a complex geological environment.

POINT N°4 As regards point N°4, I feel that the "correlation" between transmissivity and age is biased. First of all, I consider that, in that frame, resistivity should not be compared with transmissivity, but with hydraulic conductivity. The thickness of the "aquifer", which is included into transmissivity, has nothing to play with the rock's resistivity which, to my opinion, should be independent from its thickness, if the geophysical data are well inverted. Then, comparison should only be made with hydraulic conductivity (K, permeability, on Fig. 8B).

Second, as described below in additional remarks, the way to compute the hydraulic conductivity should be revised, and also better explained. Then, depending on the permeability type (generalized to the lithological body, or localized to some fissures for instance), the relationships between K and the resistivity may be challenged.

Third, as no borehole from the PdC domes is included in the data set, these formations are not used for the "correlation" between K and age. I "fear" that, including such data would completely modify the conclusions here as, to my opinion (but also as the authors suggest from the watershed budget data), such domes exhibit a rather high permeability. The authors could use the data from wells drilled in more recent domes such as the ones from the Champflore area (a few kilometers North from the study area) as proxies of the hydraulic conductivity of the PdC domes. I however "fear" that it would completely modify the relationships between formation age and hydraulic conductivity and then deny the proposed correlation.

Fourth, the impact of the lithology on hydraulic conductivity is not considered. In the

literature, and on the field in Martinique (boreholes data), the muddy debris flows (lahars, etc.) are intrinsically considered of rather low permeability, not as a result of their age, but due to their lithology.

And also, a correlation with only 3 couples of (resistivity, age) data, additionally with large uncertainty on resistivity data, is maybe a clue but, to my opinion, not a proof.

Then, the proposed explanation in the "Discussion" section (p. 12, lines 15 and followings) cannot be considered, to my opinion, as a demonstration.

Additionally, hydrochemical (major elements), and particularly a synthesis of isotopic data, would surely be of a high added value to understand groundwater flows in the studied area. Such an approach would in fact be independent from the quantitative methodologies currently developed in the paper.

CONCLUSION Then, as a conclusion and to my opinion, the proposed conceptual model of these watersheds (as well as the discussion and conclusion) will have to be revised according to the outputs of the revision of the basics of the paper. It would surely enable to much less use the conditional wording in this part of the paper, that deserves its credibility. Moreover, the watershed conceptual model will have to better consider consistency between flow lines and inferred piezometry (for instance, (i) vertical flow lines in the 9alphaBi domes, that seem to go deeper than the hyaloclastites &H are hardly understandable as hyaloclastites are considered by the authors to be of low permeability; (ii) as said above, location of all reported springs is not very consistent, low discharge springs should surely not be represented on this figure; (iii) subhorizontal arrows – flow lines – in the 2alpha formation are not consistent with the existence of strong hydraulic conductivity variations in this formation).

Then, as highlighted above in this review, below, and in the additional remarks, several of the affirmations of this section are highly conjectural to my opinion.

1. Andesitic domes To my opinion, the data from the Attila spring, and also the water budget data (if appropriately recomputed/explained) may sustain the fact that this formation can be considered as a high permeability aquifer. Then, it would be nice to use geological and geophysical data to map this aquifer, identify its surface (and underground if any) outputs, explain the hydraulic conductivity contrasts (notably with the neighbor other geological formations) that explain the location of the main spring(s), etc.. - p. 9, line 5: as said above, the way this 85% value is computed is not explained in the paper - as said above, the explanations about springs will have to be revised considering low discharge springs on one hand, and the high discharge ones such as the Attila spring. - p. 9, line 14: in hydrogeology, "source" has no meaning for a river; specific discharge must be preferred, and then the origin of this specific discharge (either high rainfall, or aquifer outflow, or both) must be discussed.

2. Andesitic lavas It is dangerous to consider this huge complex as an unique "aquifer" (see for instance p. 10, lines 17 and following: lithological continuity doesn't mean aquifer continuity; a higher resolution is needed). Again, as said above, the interpretation of the whole data set should help get into high-resolution in this aquifer, considering its lithological structure (more or less listed p. 9, lines 23 to 25), the fracturing (that could indirectly be supported by the results from the paper of Lachassagne, Léonardi, Vittecoq et al. (2011) that would be worth to be cited), and the weathering. It would help not only to deal with vague concepts such as "heterogeneity" (p. 9, line 23 for instance), but to explain the origin of this spatial and vertical variability of hydrodynamic parameters. Then, a more detailed analysis of the data set would enable to less use the conditional wording in this part of the paper.

3. Regional aquitard If this formation is really an aquitard, and if the overlying andesitic lavas are a regional aquifer largely supplied with recharge, then significant springs or outflows to the rivers should be observed at their interface. This must be better described, and conclusions issued from these observations.

4. Geothermal insight I fear that this part of the paper is highly conjectural, out of the scope of the data, and then deserves the other results of the paper. - p. 10, line 25 and

following. Be careful, the hyaloclastites are known to be of low resistivity in most areas of Martinique. To explain this low resistivity, the hypothesis of their lithology, as well as their weathering must also be considered, and not only hydrothermal weathering. Moreover, there are also similar thermal springs elsewhere in Martinique.

As a conclusion, as regards the conceptual model of the watershed, to my opinion the authors should consider the following arguments: - the lavas from the PdC domes may be a rather high permeability perched aquifer. I feel that, with a few additional arguments and explanations, this point could rather easily be demonstrated, and that their functioning as an "unique" aquifer may be supported or, at least, strongly hypothesized; - the fact that the MJ hyaloclistes constitute a regional aquitard may also surely be supported; - I fear that the other formations (PdC paleovalleys and MJ lavas) cannot be considered individually as a whole as aquifers. There is a need to get to a higher resolution to understand and describe the pervious structures and the impervious ones that they contain. Then, to my opinion, these formations are a juxtaposition of small sized aquifers and semi-pervious to aquitard formations; however, at this stage, this is an opinion, not a demonstration. With their data set, the authors may demonstrate that, or demonstrate other concepts. If I compare the results proposed in this paper with the one obtained in Mayotte (Lachassagne et al., 2014), which support the presence of a few rather high hydraulic conductivity, rather low extension (a few hundred of meters) aquifers within low to very low hydraulic conductivity formations, I would consider that the MJ aquifers may be smaller (and of lower hydraulic conductivity) than those or Mayotte. In Mayotte, the determinisms of the hydraulic conductivity was described and demonstrated (for instance in lavas = discrete clinker layers and thicker cooling fractures layers), the present paper should help to provide similar observations in the Martinique MJ formations; - then, the "correlation" between permeability and age, and, beyond that, the interpretation that tectonic fracturing mostly governs the hydraulic conductivity of these rocks is, to my opinion, with the available arguments and data, highly conjectural. Moreover, the authors should not only consider the benefits of earthquakes on hydraulic conductivity (cf. Lachassagne, Léonardi et al. (2011)), but

also the impacts of the clogging processes that fast follow the permeability increase due to the earthquake (see for instance, the paper from Lachassagne, Wyns et al. (2011) that summarizes all these issues).

ADDITIONAL REMARKS Some additional remarks or points to be addressed: - Fig1: a few French words are remaining ("élevée", "faible")

- Fig 1 & 2: the 2 maps contain probably too much data and are then hard to read. The spring symbol and/or color should be changed as blue over blue is not visible. The AEM data should be presented with dots (location of data) and not lines.

- p. 3, line 27: this is not annual temperatures that vary from 18°C to 32°C in Fort-de-France. Please rephrase

- p. 3, line 31 and following, p. 6, etc.: maybe the wording of "surface water intake" or "SW catchwork" should be preferred to "dam" in the present case

- p. 4, line 24: domes?

- p. 4, line 26: flow out?

- p. 5, line 13 and following: Hydraulic conductivity The authors highlight that the hydraulic conductivity was computed by dividing transmissivity by the height of the saturated screen interval. This is a rather rough approach. They could describe to which kind of lithology it refers, as well as to which kind of permeability (interstices, thermal or flow cracking, clinker layers, tectonics, etc.) and, where necessary, weathering. In fact, in such geological formations, most of the discharge/hydraulic conductivity of a given borehole can come from very narrow intervals. Then, dividing the transmissivity of the well by all the screened interval may not be relevant. Then, a detailed description of the geological logs, and their correlation with water inflows should be performed and some examples proposed as well as statistics. Such an approach was for instance performed by Lachassagne et al. (2014) (see Figs 4 and 5 of their paper). Instead, these values of hydraulic conductivity should be considered only as orders of magnitude, and

I would recommend to not cite them in the paper, and keep transmissivity values as orders of magnitude of the relative productivity of the different geological formations.

- p. 5, line 17 and following: piezometry As highlighted by the authors, these data show that the piezometric level are shallow, and that there is no identified "basal aquifer". Nevertheless, a higher resolution interpretation of these piezometric data could surely provide additional information, notably on the extension of most aquifers. For instance, by zooming on local data from Fig. 3B, the authors could also surely discuss about piezometric gradients at the local scale of a given aquifer; this local gradient is surely much less than the apparent mean 4% "gradient" visible on Fig. 3B; this latest surely reflects the topographic slope. From these data, I suspect that most aquifers have a very small lateral extension (a few hundreds of meters at the max. This clue resulting from the piezometric data interpretation should be mentioned in the paper. It may help to understand groundwater flows at the watershed scale, and refine the proposed conceptual model. Again, to complete my previous remark about springs, the discharge of the springs should be indicated on Fig. 3, for instance with an appropriate symbol/legend.

- p5, line 24: confined

- Fig. 4 : effective rainfall is not monitored; prefer computed? Piezometric level in m (and not mamsl as a shift was made to enable their drawing on the same graph). There is surely a mistake in the legend as the Case Pilote borehole is cited in the caption, but not in the legend. No influence of pumping on some data? Notably Case Navire 165 in 12/2006, 06/2009?

- p. 5, line 27: mean annual specific discharge?

- Fig. 5: liter should be abbreviated "L", and not "l"

- Line 16, p. 6: "...2012. It allows..."

- Arnaud et al. (2013) is not listed in the references

- p. 8, line 28: characterizes

- p. 8, line 29: highlights

- p. 10, line 10: it's not really "underlying" aquifers but rather neighbor ones

- p. 10, line 11: "a significant"

- Fig. 7: there is no legend for the vertical dotted lines. Also lithological contact like the subhorizontal ones?

- Fig. 6 & 7: springs must be located on the profiles: C1 = 1 spring, C4 at least4 springs, C5 1 only (and not 2). Moreover, springs should be plotted at their real elevation to show the correlation with the geophysical structures. Topographic artefact are also visible ; they could be discussed.

- p. 10, line 20: what means computation "of high flowrates"? (i) computation (moreover not presented in the paper) is not a proof of sustainable discharge, (ii) high flowrate at a given well doesn't mean high natural flow in the aquifer, (iii) pumping may completely change the groundwater age, etc.. Then, as it is written, the demonstration in this section is not convincing.

Additional references cited in the review:

Lachassagne, P. (2003). SYNDICAT DES COMMUNES DE LA COTE CARAÏBE NORD-OUEST - CONSEIL GENERAL DE LA MARTINIQUE. Source Attila (Commune du Morne Vert, Martinique). Délimitation des périmètres de protection du captage et détermination des prescriptions associées (in French – registered hydrogeologist report for groundwater protection zones delineation) PL - AHAHP - 03 MTQ 03, 17 p.

Lachassagne, P., Aunay, B., Frissant, N., Guilbert, M., Malard, A. (2014). High-resolution conceptual hydrogeological model of complex basaltic volcanic islands. A Mayotte, Comoros, case study Terra Nova, Vol. 26, N°4, PP. 307-321, DOI 10.1111/ter.12102

Lachassagne, P., Léonardi, V., Vittecoq, B., Henriot, A. (2011). Interpretation of the piezometric fluctuations and precursors associated with the November 29, 2007, magnitude 7.4 earthquake in Martinique (Lesser Antilles). Comptes-Rendus Geosciences, 343 (2011) 760–776

Lachassagne, P., Wyns, R., Dewandel, B. (2011). The fracture permeability of hard rock aquifers is due neither to tectonics, nor to unloading, but to weathering processes, Terra Nova, 23, 145-161

---

## Referee Comment (RC1) · Tatiana Izquierdo (Referee) · 27 Feb 2019

General comments: The manuscript by Vittecoq et al. presents a hydrogeological conceptual model of andesitic watersheds by combining hydrologic, hydrogeologic, geologic and geophysical methods. The study has been carried out in the island of Martinique and includes new insights about the hydrogeology of andesitic islands that has been far less studied than basaltic islands. It is, therefore, an interesting manuscript that should be considered for publication in Hydrology and Earth System Sciences.

Specific comments

- The authors have included the thermal springs as part of the hydrogeological model, however, more specific information is needed to fully understand them and their role in the hydrogeological conceptual model.

- Another table with supplementary information is needed with the available data for the analyzed springs (location, geology, discharge, seasonality,...)

- Page 5, line 24: you mention that piezometer 3 characterizes a confined aquifer (figure 4) however according to the cross-sections interpretation there is no unit to confine the aquifer. The supplementary information table says it crosses 1a unit although that unit does not appear in table 1. Could you please include a more detail explanation? Maybe including the boreholes location in the cross-sections?

- The water balance section needs to improve the methodology of the obtained values. Temporal series used for rainfall and evapotranspiration values should be specified and the method used for the evapotranspiration calculation should be included or at least a reference for them. How did you estimate GwR and the runoff values?

- Page 8, line 12: "... a specific analysis was conducted in the springs." Which one? Please provide a brief explanation of the methodology.

Minor corrections have been included in the annotated pdf file.

Please also note the supplement to this comment:
https://www.hydrol-earth-syst-sci-discuss.net/hess-2018-637/hess-2018-637-RC1-supplement.pdf

**Supplement:**

[revised manuscript text omitted]
⁻¹) Nbr. of boreholes | range | Average and std. | Hydraulic conductivity (m s⁻¹) range | Average and std. | Water electrical conductivity µS cm⁻¹ Nbr. Of springs | Springs | Rivers |
|---|---|---|---|---|---|---|---|---|---|---|---|---|---|---|
| ⁹αbi | ⁹αbiᴰ - Andesitic domes | 0.9 | > 500 | Springs : 150 - 500 | Upper major perched aquifer | highly fissured/fractured | - | - | - | - | - | 6 | 106 - 156 | 111 |
| | ⁹αbiᶜ - andesitic lava flows | 0.9 | 10 - 100 | Springs: 100 - 180 Boreholes: 50 - 100 | Upper minor perched aquifer | highly fissured/fractured | - | - | - | - | - | - | - | 124 - 142 |
| ⁸ρα | Andesitic and dacitic domes and lava flows | 1 | 10 - 100 | 70 - 120 | Minor perched aquifer | Fissured/fractured | - | - | - | - | - | - | - | - |
| ⁶B/C | Breccias, debris flow from the first phase of construction of the old Carbet | 2 | 100 - 200 | 20 - 40 | Minor perched aquifer | Heterogeneous Locally aquitard (cemented breccias) | 3 | 2.3 10⁻⁴ to 3.7 10⁻⁴ | 2.9 10⁻⁴ std: 5.8 10⁻⁵ | 6.7 10⁻⁶ to 1.7 10⁻⁵ | 1.1 10⁻⁵ std: 4.6 10⁻⁶ | - | - | - |
| ²α | Andesitic lava flows | 2.2 - 2.8 | 100 - 300 | - 2αᴹ (massive parts): 30 - 70 - 2αᶠ (fissured and fractured parts): 15 - 30 - 2αᵇ (brecchias and autoclastics parts): 10 - 15 | Major aquifer | Heteregeneous | 9 | 1.0 10⁻⁵ to 7.0 10⁻⁴ | 3.3 10⁻⁴ std: 2.4 10⁻⁴ | 2.3 10⁻⁷ to 1.8 10⁻⁵ | 9.0 10⁻⁶ std: 6.4 10⁻⁶ | 7 | 50 - 200 | 91 - 120 |
| ¹βol | basaltic lavas | 4 - 5.5 | 100 - 300 | 16 - 20 | Major aquifer | Fissured/fractured | 7 | 4.4 10⁻⁴ to 1.3 10⁻³ | 7.7 10⁻⁴ std: 2.9 10⁻⁴ | 8.8 10⁻⁶ to 3.3 10⁻⁵ | 2.0 10⁻⁵ std: 7.5 10⁻⁶ | - | - | 91 - 98 |
| ¹H | hyaloclastites | 4 - 5.5 | > 200 | 6 - 10 | Regional aquitard | very low permeable formation | - | - | - | - | - | - | - | - |

**Table 1: Geological, geophysical and hydrogeological characteristics of the main aquifer and aquitard formations.**

---

## Editor Comment (EC1) · Gerrit H. de Rooij (Editor) · 4 Mar 2019

I thank Dr. Lachassagne for his very detailed comment. For clarity I point out that this comment was volunteered, and not one of the reviews requested by me.

The comment reveals a thorough knowledge of the geomorphology and hydrogeology of Martinique, and I am sure the authors can greatly benefit from the insights provided in the comment.

As we move forward with the discussion of this paper, I would like to remind the discussion partners about the international readership of HESS. The discussion paper

examines how new data collection methods can be used to develop hydrogeological models of andesitic watersheds, and uses Martinique as a case study. Dr. Lachassagne's comment shows that the authors chose their test site wisely: there is a wealth of geological information available that can be used to test their approach.

I would suggest as we move forward not to deviate too much from the current focus of the paper: testing the ability of new geophysical tools to support hydrogeological model development. I gather from the discussion so far that the available data (old and new) would permit a refinement of the current geological map of Martinique and the development of a detailed model of the hydrogeology of the island. These are undoubtedly worthy endeavours in their own right, but they are of limited interest to the majority of the HESS readership.

I therefore encourage the authors to consider Dr. Lachassagne's comment in light of the stated objectives of their study. I am confident the comment will help them clarify, refine, and perhaps even improve elements of their analysis.

Gerrit de Rooij

HESS Editor

---

## Referee Comment (RC2) · Anonymous Referee #2 · 5 Mar 2019

This manuscript presents a multidisciplinary study including geological, hydrological and geophysical data for characterizing andesitic watersheds located in Martinique Island, in the Lesser Antilles Archipelago. As motivated by the authors, such specific hydrogeological setting has taken less attention in the literature than, e.g., in basaltic islands. For this aspect, I think this study can be published in the journal HESS.

The manuscript is clear and well organized. However, because the geophysical helicopter-borne TDEM method is presented as a new key element for constraining the hydrogeological model, I think the authors should provide few more details about their data processing workflow. Please consider the following comments:

[Figure]

1- Since the clearance is derived from the DEM model and the DGPS altitude, why is it needed to invert for the altitude of the transmitter?

2- The principle and the function of the SVD filter method must be explained, even succinctly. Indeed, SVD is a very general concept used in many other contexts.

3- Similarly, some details should be provided to explain why a trapezoidal filter is used to filter the TDEM data (e.g., because of increasing lateral footprint with time windows etc..).

4- How important is the trapezoidal lateral filter in practice? Does it smooth only the noise or also some 2D/3D effects in the data? Or, are local 2D/3D effects considered as noise in this study?

5- Is it worth and safe (with the price and risk of minimizing an optimum function involving a large number of non-correlated data) to use an expensive SCI algorithm to invert profiles, which barely overlap each other (400 m is a large distance with respect to the TDEM lateral footprint for most of the time windows channels, making the data set/maps highly under-sampled in the crossline direction)?

6- A lateral (trapezoidal) smoothing filter is applied to the data before inversion. In addition, lateral smoothness constraints are applied in the inversion. It must also be recalled that, by definition, the 1D forward modelling algorithm used for the data inversion assumes a layered earth (with no lateral variations and a flat ground surface).

These three layers of constraints/assumptions basically force flatness in the results, at the scale of the TDEM lateral footprints. This approach could erroneously give the impression of flatness even if it is not the case in the reality, at a local scale. According to this study, it seems actually sufficient to properly characterize formations at the watershed scale. However, I think it is fair to recall that such a workflow inherently loses a substantial part of lateral details, which is present in the recorded TDEM data. To sum up, according to basic signal processing criterions, the overall acquisition + filtering + inversion approach presented in this study does not fully exploit the potential of the helicopter-borne TDEM method in term of lateral resolution. This makes the term "high-resolution" in the title maybe not appropriate for the presented approach; from the geophysical perspective. This is not really a criticism as the acquisition of TDEM data is difficult in practice, in such a mountainous area, and because 2D/3D imaging algorithms are still hardly available for processing such large data sets. However, I think it is important to highlight these limitations anyway, as we must expect further developments concerning the TDEM method, with a spatial coverage reaching the Nyquist's sampling theorem condition and more robust processing approaches.

Best regards

---

## Author Comment (AC1) · 8 Mar 2019

We thanks Tatiana Izquierdo Labraca for her consideration for our work and for agreeing reviewing our manuscript for publication in Hydrology and Earth System Sciences. Her relevant comments will clearly help to improve the manuscript.

Responses to specific comments:

C1: The authors have included the thermal springs as part of the hydrogeological model; however, more specific information is needed to fully understand them and their role in the hydrogeological conceptual model.

R1: In the present form of the manuscript, the two thermal springs are presented in chapter 2.3.2 and their hydrogeological functioning is briefly described in chapter 5.6. More details will be added in the revised manuscript accordingly to the information we have regarding those thermal springs. Information about the geological and hydrogeological local settings of those two springs will be added in the chapter 2.3.2 and their role in the hydrogeological conceptual model will be clarified in chapter 5.6. We also consider to add a second figure to the conceptual model in order to show, from another angle, a cross section from north to south in the Case Navire River, and through the two thermal springs.

C2: Another table with supplementary information is needed with the available data for the analyzed springs (location, geology, discharge, seasonality, etc.)

R2: As recommended, a table with spring database will be added.

C3: Page 5, line 24: you mention that piezometer 3 characterizes a confined aquifer (figure 4) however according to the cross-sections interpretation there is no unit to confine the aquifer. The supplementary information table says it crosses 1a unit although that unit does not appear in table 1. Could you please include a more detail explanation? Maybe including the boreholes location in the cross-sections?

R3: The geological log of piezometer 3 shows that andesitic lavas are covered by 20 m of argilized material, which may correspond to clayey scree or clay alluvium. This clayed horizon confines the aquifer in this valley. The title of the column "alluvial depth" will be modified to include information about the superficial formation such as clayed material, highly weathered lavas or argilized alluvium.

$1\alpha$ and $1\beta$ol are both from the first volcanic phase of the Morne Jacob volcano. Table 1 (and also figure 8) already include $1\alpha$ and $1\beta$ol characteristics. We will correct the code of table 1 by "$1\alpha$ and $1\beta$ol", and columns $1\alpha$ and $1\beta$ol will be aggregate in the supplement boreholes table 1 for clarity.

The drilling is located near the 8000 m abscissa on the C5 cross section. We didn't plot this borehole on the cross section as the flightline was flown over the valley flank and not over the bottom of the valley. Considering the lateral extension of lavas, the $1\alpha/1\beta$ol is not imaged on the flank.

C4: The water balance section needs to improve the methodology of the obtained values. Temporal series used for rainfall and evapotranspiration values should be specified and the method used for the evapotranspiration calculation should be included or at least a reference for them. How did you estimate GwR and the runoff values?

R4: Water balance are voluntarily presented at the beginning of the paper, in the chapter on general knowledge, as we synthesize existing data from different organizations (Meteorological agency, water office, ecological Ministry...) and existing results from rainfall/flow modelization. This will be better introduced and clarified in the introduction of this chapter. Some details and references have also to be added regarding the methodology used to calculate the different hydrological terms of the water budget (Vittecoq et al., 2007, Vittecoq et al., 2010, Arnaud et al., 2014, Stollsteiner et Taïlamé, 2017, data from Meteorological agency and the Ministry of ecology, etc.), in addition to uncertainties of the main measurements.

RET are spatialized data from Arnaud et al., 2014, following the methodology detailed in Vittecoq et al., 2010. GwR/runoff ratio have been calculated by Vittecoq et al., 2007 (over the period 1987-1990) and Stollsteiner et Tailamé, 2017 (over the period 2008-2015) using global model with reservoir Tempo and Gardenia ($^{©}$ BRGM).

C5: Page 8, line 12: a specific analysis was conducted in the springs." Which one? Please provide a brief explanation of the methodology.

R5: Resistivity were extracted manually from the 3D resistivity model. Cells situated upstream springs were selected. This extraction was done for all of the 24 springs of the database (this database will be include in the revised version) and the sentence modified for clarity.

Responses to minor corrections (in RC1-Supplement.pdf):

R6: page 1, line 4: Yes, we mean overexploitation. This will be modified in the revised version of the manuscript.

R7: page 2, line 9: The reference Izquierdo, 2014 was added.

R8: page 3, line 20: Explanations about the objective "age is not the key factor controlling hydraulic conductivity" are given below.

Thanks to the interpretation of the geological, geophysical and hydrogeological data, we highlight, for the present study (i.e. the watersheds and the three studied aquifers, within the interval 10-100 ohm.m and within a range of 0.9 to 5.5 Ma) that (1) the older the formation, the lower its resistivity and (2) the older the formation, the higher its transmissivity. This last result is also consistent considering the results of Vittecoq et al., (2015) obtained on an older aquifer (15 Ma) on Martinique Island, with higher hydraulic conductivity and lower resistivity than the ones observed in the present study.

Consequently, unlike hot spot basaltic islands, hydraulic conductivity of the studied aquifers (subduction zone andesitic volcanism) does not decrease with age. On the contrary, our results show an increase with age. Nevertheless, time itself is not the activating factor and only few geological processes can cause an enhancement of permeability. Given (1) the tectonic and seismic context of the subduction zone, (2) the fact that earthquakes are known for increasing permeability (e.g. Rojstaczer et al., 1995, Ingebritsen et al., 2008) and (3) the fact that earthquake induced modification of permeability have been observed in Martinique (Lachassagne et al., 2011), we interpret the observed permeability increase as the consequence of earthquake tectonic fracturing.

It should be noted that this process must be superimposed with others geological process occurring on the same time and acting to reduce permeability, such as weathering process or hydrothermal argilisation. Intense hydrothermal weathering and fracture

sealing by geothermal fluid could indeed create an impermeable caprock with very low resistivity (e.g. Browne, 1970, Simmons et Browne, 1990, Dobson et al., 2003), as supposed for hyaloclastites 1H.

We rephrased the last objective of our paper in that way: "(v) strengthen the hypothesis of Vittecoq et al., 2015 that, in contrast with the basaltic islands, hydraulic conductivity may increase with age in andesitic-type volcanic island."

R9: page 3, line 28, 29: we agree, we will add that the rainfall map is included in figure 1B.

R10: page 3, line 30 and 32: This is a translation mistake. We mean urban community / urban area. Fort-de-France is part of CACEM urban community, which also includes the cities of Schœlcher, Saint-Joseph and Le Lamentin.

R11 (page 6, line 9): Piton Lacroix will be added in figure 1.

R12 (page 9, line 30): We agree, we mean limit of the extension of andesitic lavas.

R13 (page 10, line 4): The sentence was rephrased: "A part of effective rainfall (18%-40% depending on the watershed as shown in fig. 5) deeply infiltrates through the fracture and in the rooting of andesitic domes 9$\alpha$bi".

R14 (page 10, line 26): the sentence was rephrased, as the second hypothesis is the most likely: "The very low resistivity (6-10 ohm m) of Hyaloclastites 1H cannot correspond to actual salt water intrusion as they are now situated higher above sea level. Their very low resistivity could result from hydrothermal weathering (e.g. Browne, 1970, Simmons et Browne, 1990). These low resistivity hyaloclastites would thus be an evidence of a hydro-thermalized caprock of an underneath geothermal system".

R15 (Fig 1): Yes, we agree. We replaced the number by the associated name in the figure.

R16 (fig 1): all the boreholes and springs shown on this figure are used in our

paper. Boreholes and springs outside the watershed are in the same geological/hydrogeological formations and are necessary as proxy for statistical analysis.

R17 (fig 2): Some flight lines outside the watershed will be removed. The color of the flight lines will also be changed, for clarity.

R18 (fig 4): Piezometric levels increase since 2009/2010 are linked to the variation of effective rainfall (cf. table below). The years 2006 to 2009 have suffered a rainfall deficit, while the year 2010 and 2011 where excess and followed by two normal years.

Difference with the average annual effective rainfall: 2005: +35%, 2006: -19%, 2007: -17%, 2008: +1%, 2009: -26%, 2010: +30%, 2011: +60%, 2012: +4%, 2013: +5%, 2014: -29%, 2015: -34%

R19 (figure 7): Yes, the elongated rectangle down is a borehole. The figure will be modified to add this information, likewise the dotted lines that are actually faults. Orientations of C4 and C5 will also be added (from north to south).

R20 (figure 8): Andesitic and basaltic lavas "4-5.5 Ma" correspond to $1\alpha$ and $1\beta$ol (see also R3)

R21 (figure 9 - Thermal springs): See R1 for our response.

R22 (figure 9 – Downstream the water tank): Yes we agree, diffuse springs are observed and the river drains the aquifer in this part of the valley. We will add a specific mark.

R23 (figure 9 - Title): We agree, and we proposed "Hydrogeological conceptual model, at watershed scale, of an andesitic system in a context of subduction zones".

R24 (table 1 - $1\beta$ol): Yes we agree, we consider $1\beta$ol as a major aquifer at the scale of the island (and also $1\alpha$, cf. R3). Figure 2 also show that $1\alpha$ and $1\beta$ol are found mainly at the East of Alma and Case Navire watersheds, but the angle of view of the conceptual model does not allow showing them. Given that (1) $2\alpha$, $1\alpha$ and $1\beta$ol follow

each other in time (2) they are all aquifer (keeping in mind that they are heterogeneous with permeable and impermeable facies, cf. chapter 5.2, lines 22-25), and (3) this is a conceptual model, we propose to modify the legend with generic items: andesitic domes, lavas, debris flows, andesitic aquifer and hyaloclastites.

References:

Arnaud, L., Lanini, S.: Impact du changement climatique sur les ressources en eau de Martinique. Openfile BRGM Report RP-62676-FR, 2014.

Browne, P.R.L.: Hydrothermal alteration as an aid in investigating geothermal fields. Geothermics, Volume 2, Part 1, pages 564-570, ISSN 0375-6505. https://doi.org/10.1016/0375-6505(70)90057-X, 1970.

Dobson, P.F., Kneafsey, T.J., Hulenb, J., Simmons, A.: Porosity, permeability, and fluid flow in the Yellowstone geothermal system, Wyoming J. Volcanol. Geotherm. Res., 123 (3–4, 1), pp. 313-324, 2003.

Ingebritsen, S.E., Sanford, W.E., Neuzil, C.E.: Groundwater in Geologic Processes. Cambridge University Press, 365 pp., ISBN 0-521-49608,1998 Second Edition, 2006.

Izquierdo, T.: Conceptual Hydrogeological Model and Aquifer System. Classification of a Small Volcanic Island (La Gomera; Canary Islands). CATENA 114, 119-128, 2014.

Johnston J.M., Pellerin L., and Hohmann G.W.: Evaluation of electromagnetic methods for geothermal reservoir detection exploration. Geothermal Resources Council Transactions, vol. 16, pp. 241-245, 1992.

Lachassagne, P., Léonardi, V., Vittecoq, B., Henriot, A.: Interpretation of the piezometric fluctuations and precursors associated with the November 29, 2007, magnitude 7.4 earthquake in Martinique (Lesser Antilles). Comptes-Rendus Geosciences 343, 760–776, 2011.

Rojstaczer, S.A., Wolf, S.C., Michel, R.L.: Permeability enhancement in the shallow

crust as a cause of earthquake-induced hydrological changes. Nature 373(6511):237-239. DOI: 10.1038/373237a0, 1995.

Stollsteiner, P., Taïlamé, A.L.: Détermination des seuils de vigilance des niveaux d'eau souterraine en Martinique. Openfile BRGM Report RP-66058-FR, 2017.

Simmons, S.F., and Browne, P.R.L.: A three dimensional model of the distribution of hydrothermal alteration mineral within the Broadlands-Ohaaki geothermal field: Proc. 12th New Zealand Geothermal workshop, 1990.

Vittecoq, B., Lachassagne, P., Lanini, S., Ladouche, B., Marechal, J.C., and Petit, V.: Élaboration d'un système d'information sur les eaux souterraines de la Martinique : identification et caractérisations quantitatives. Openfile BRGM Report RP-55099-FR, 2007.

Vittecoq, B., Lachassagne, P., Lanini, S. and Maréchal, J.C.: Evaluation des ressources en eau de la Martinique : calcul spatialisé de la pluie efficace et validation à l'échelle du bassin-versant. Revue des Sciences de l'Eau 23 (4), 361–373, 2010.

---

## Author Comment (AC2) · 12 Mar 2019

We thank the referee #2 for reviewing our manuscript for publication in Hydrology and Earth System Sciences. Her comments will help to improve and clarify the manuscript.

Responses to comments:

C1: "Since the clearance is derived from the DEM model and the DGPS altitude, why is it needed to invert for the altitude of the transmitter?"

R1: Over rugged terrain, the DEM model allows obtaining a more consistent altitude than the one derived from laser altimeter. However, DGPS may suffer from inaccuracies mainly on the z-axis and the resolution of the DEM may not suit to the measurement at some places. Thus, we decided to invert for the altitude even if the use of the DEM allows inverting for the altitude after more iterations than what is done commonly.

C2: "The principle and the function of the SVD filter method must be explained, even succinctly. Indeed, SVD is a very general concept used in many other contexts"

R2: Of course, SVD is used in many other contexts. The way we use the SVD is fully described in Reninger et al., 2011 (cited in the manuscript). We will add some information on the procedure in the text (page 7, line 18): The SVD allows explaining a dataset with only few components, each data being a linear combination of these components. Thanks to this decomposition we are able to identify and remove several types of noise, making the processing less time consuming and subjective.

C3: "Similarly, some details should be provided to explain why a trapezoidal filter is used to filter the TDEM data (e.g., because of increasing lateral footprint with time windows etc.)."

R3: a sentence will be added about the trapezoid stacking: (page 7, line 19): The trapezoid shape is consistent with the increasing of the footprint of the EM method with time.

C4: "How important is the trapezoidal lateral filter in practice? Does it smooth only the noise or also some 2D/3D effects in the data? Or, are local 2D/3D effects considered as noise in this study?"

R4: The trapezoid stacking is adapted to the noise level at each measurement, unnoisy windows are not stacked in order to unalter the lateral resolution. The detailed methodology is presented in Reninger et al., 2018 (cf. reference in the manuscript). A new sentence will be added to clarify in the text (page 7, line 19): Thanks to this filter we try to retrieve the noisy windows, which are unusable otherwise.

C5: "Is it worth and safe (with the price and risk of minimizing an optimum function

involving a large number of non-correlated data) to use an expensive SCI algorithm to invert profiles, which barely overlap each other (400 m is a large distance with respect to the TDEM lateral footprint for most of the time windows channels, making the data set/maps highly under-sampled in the crossline direction)?"

R5: The SCI algorithm used in the Aarhus workbench proved its robustness in many published studies, even when flightlines are acquired mainly in one direction, and is become a standard. As shown in Figures, The SCi also allowed making good use the intersections between lines. Moreover, the inversion was set with weak lateral constrains. A sentence will be added in the text (page 7, line 25): weak constrains were applied for this study in order to limit the smoothing of the inversion procedure.

C6: "A lateral (trapezoidal) smoothing filter is applied to the data before inversion. In addition, lateral smoothness constraints are applied in the inversion. It must also be recalled that, by definition, the 1D forward modelling algorithm used for the data inversion assumes a layered earth (with no lateral variations and a flat ground surface)."

R6: Trapezoid stacking is applied on noisy windows only in order to retrieve windows, which are unusable otherwise. Unnoisy windows are not stacked, the lateral resolution is therefore unaltered. Moreover, weak lateral constrains were applied. Our processing tried to keep as much lateral resolution as possible (Reninger et al, 2018), this will be added to the text (page 7, line 28): The aim of the applied processing was to keep as much resolution as possible (Reninger et al, 2018). Concerning 1D model, they are described page 7, line 22-23.

R7: Given the number of data used and the size of the study area, we think that it is rather fair to use the words "high-resolution". Moreover, as it is noticed by the reviewer, it is not possible to obtain results of better resolution at this time. Nevertheless, the title of the paper may be modified in this way: "Hydrogeological conceptual model of andesitic watersheds revealed by helicopter-borne electromagnetic geophysics".

637, 2019.

---

## Editor Comment (EC2) · Gerrit H. de Rooij (Editor) · 29 Mar 2019

Dear authors,

The review comments that were provided were detailed and to the point. Your replies to them were thorough and showed that you have a clear view of the way you want to address these comments. Overall, the reviewers view the paper positively, and I share that view.

I therefore kindly request you to revise the paper along the lines you have indicated in your replies. I believe this will mainly comprise adding background/methodological

information and clarifying sections of the text, but no major additional analysis, calculations, etc. I therefore believe a minor revision will suffice. I may ask a brief review of the revision, but perhaps I can rely on my own judgement.

Yours sincerely,

Gerrit de Rooij

Editor

---

## Author Comment (AC3) · 29 Mar 2019

We firstly want to thanks Patrick Lachassagne for his interest for our work and the ensuing extended short comments. Remarks and comments will help to improve the manuscript in the next step of the review process, following the recommendation of the handling editor.

The main objectives have been clearly identified and summarized by P. Lachassagne: (i) Improving knowledge of the hydrogeological functioning of an andesitic-type volcanism in a subduction zone (this kind of context being much less studied and documented) (ii) with helicopter-borne geophysics (which is clearly not so common in vol-

canic context), (iii) bringing new correlations between age, transmissivity, permeability and resistivity and (iv) proposing a conceptual model and hypothesis about hydrogeological specificities of andesitic watersheds.

We below respond to the main topics, which include general, minor and additional comments.

Concerning geological and geophysical remarks, as pointed by P. Lachassagne, the existing geological map is of high quality and high resolution, notably in this area of the island. That is the reason why we choose those watersheds in order to test how helicopter-borne geophysics can develop hydrogeological models of andesitic watersheds. Considering aims, scope and readership of HESS journal, and the multidisciplinary angle of our paper, the challenge was to synthesize the essential data of each discipline (geology, geophysics, hydrogeology), necessary to support our results and then our hypotheses and discussion. Our results show the advantage of electromagnetic data to highlight the thicknesses, lateral extension and morphology of the different geological formations. Correlations with geological map and geological log from boreholes allow showing the interest of electromagnetic data and how this kind of survey can be used in areas where there is no geological map in order to better understand hydrogeological functioning.

As explain in chapter 3.1, the geophysical survey was conducted along the N-S direction with a resolution of an EM sounding every 30 m along flight lines, and with a 400 m line spacing (ie. One flight-line every 400 m in the E-O direction). As reminded by P. Lachassagne, and as explained in the introduction section and in paragraph 5.2, andesitic formation are classically marked by a high lateral and vertical geological variability whether within each formation or between successive formations. Consequently, our approach focus on interpreting geophysical data along the flight lines (rather than in the interpolated 3D block) precisely because of the limited lateral extension of permeable/impermeable facies within each formation. In most case, there is a good consistency between geophysical cross sections and the geological map. However, this

consistency can only be demonstrated after conducting a work as the one presented here, as geophysical data interpretation is an iterative process depending on calibration data. Comparison with the few geophysical ERT available have also been done but, (1) as they only concern a small part of the Fond Lahaye valley, and (2) as they are not enough closed to flight lines, their interest with respect to our working scale and the objective of the paper is limited. Correlations between the two geophysical methods is interesting for borehole sitting but it is not the scope of our paper and of HESS journal.

On the contrary to what is hinted in the short comment, we clearly do not consider existing hydrogeological conceptual models of basaltic islands, cited and described in the introduction of our manuscript (lines 9 to 13, 19 and 20 and 28-29), as "old low resolution conceptual models" by P. Lachassagne. Each cited study contributes to the building of general knowledge, according to the methodologies used (geological and hydrogeological survey, numerical modelling, geophysics, hydrochemistry, etc.) and one challenge is the necessity to better constrain the internal structure of the volcanic edifices. Lachassagne et al., 2014 present what he called the nowadays needed and high-resolution model for complex volcanic islands, considering that groundwater mainly flows through few high hydraulic conductivity and low extension formations as a succession of perched aquifer. For instance Ecker, 1976 already expose that, in basaltic islands, groundwater flows were constrained between permeable and impermeable zones, these compartment being irregular in volume, shape and structure (resulting from many factors: initial relief, erosional phases between eruptions, morphology of lava flows, weathering, erosion, fracturing, tectonic movements, etc.) forming a kind of tortuous aquifer. Indeed, it is well known that volcanic formations exhibit, by definition (e.g. Ingebritsen et al., 2006), extreme spatial variability or heterogeneity, both among geologic units and within particular units, with large variation from core scale to regional scale, permeability being, especially in volcanic environment, a scale-dependent property.

Concerning the water balance remarks, In our approach, water balance are voluntarily presented at the beginning of the paper, in the chapter on general knowledge, as we synthesize existing data from different organizations (Meteorological agency, water office, ecological Ministry...) and existing results from rainfall/flow modelization. This will be better introduced and clarified in the introduction of this chapter. Details and references will be added regarding the methodology used to calculate the different hydrological terms of the water budget (Vittecoq et al., 2007, Vittecoq et al., 2010, Arnaud et Lanini, 2014, Stollsteiner et Taïlamé, 2017, data from Meteorological agency and the Ministry of ecology, etc.), in addition to uncertainties of the main measurements.

In terms of perspectives concerning water balance calculation, the national meteorological agency is working on the calibration of their new meteorological radar in order to provide better rainfall data for future research programs (allowing in the future detailed water balance calculation at a smaller scale, spatially and temporally). In parallel, the Ministry of Ecology is installing gauging stations and the water office tried to convince the operators to equip the water supply dams with volumetric meters.

Concerning the specific case of the Attila spring, it is hazardous to suggest that it could have a significant impact on the functioning of the studied watershed. "Pitons du Carbet" domes are a set of andesitic and dacitic domes covering an area of 20 km$^2$ which have intruded between 0.9 and 0.3 Ma (Germa et al., 2011). This ensemble is the highest point of the central part of the island, with five main domes exceeding 1000 m and smaller peripheral domes. Given their topographical position, a dozen of hydrological watershed originate within this ensemble. The topographic catchment of the upper Mitan river (Where Attila spring is situated downstream) is therefore not contiguous with the watersheds we studied, as they are separated by three watersheds. As exposed by Lachassagne, 2003, both the flowrate of the spring and the recharge upstream were very badly constrained and the spring flowrate used by Lachassagne, 2003 is an estimation. Using the same calculation methodology that Lachassagne (2003), we recalculate the water balance of this watershed using new recharge modelization (Arnaud et al., 2013), infiltration/runoff ratio provided by Vittecoq et al., 2007

and Stollsteiner et Taïlamé, 2017, and the monitoring of water volume from the spring (Average value 6 l/s – max 10 l/s, but considering that the outflow is not monitored, 30% should be added). The conclusion is that the flowrate of this spring can largely be explained by the effective rainfall in the hydrologic watershed, in agreement with the geological structure of a dome, and confirm that this spring is not a regional outflow.

Concerning the correlation between transmissivity and age remarks, Fig 8B present both transmissivity and hydraulic conductivity values. As aquifer thickness are equivalent from one borehole to another, transmissivity and hydraulic conductivity show the same trend. Indeed, in this kind of fissured/fractured aquifer, water mainly flows into fractures, but, at the watershed scale, macroscopic hydrodynamic parameters values are of major interest than microscopic water velocity measurements at the fissure scale. In addition, resistivity data from the smooth inversion is obtained at macroscopic scale (one resistivity data for each of the 23 layers from 0 to 200 m depth).

Our results and correlation, (1) the older the formation, the lower its resistivity and (2) the older the formation, the higher its transmissivity, are relevant for the three studied aquifers, within the interval 10-100 ohm m and within a range of 0.9 to 5.5 Ma. The inferior limit (10 ohm m) is very important because below this value the correlations between resistivity and geological nature or hydrogeological property are not univocal. Several authors have put in evidence that this may correspond to high permeability formations saturated with saltwater (old confined water or seawater intrusion) or to impermeable clays. This clays resulting from meteorological or hydrothermal weathering process. The superior limit (100 ohm m) is also to consider, as there are no boreholes on the studied watershed with transmissivity / hydraulic conductivity values crossing formation with resistivity value higher than 100 ohm m.

We confirm that we consider that the domes exhibit rather high hydraulic conductivity, as described in the chapter 2.3 and 5.1 ("upper major perched aquifer. . . highly fissured and fractured, conferring a great hydraulic conductivity to this aquifer"). Champflor Borehole (50 m depth), situated 6 km north from the northern part of our studied water-
shed (piton Lacroix) and drilled in $8\alpha$ dacitic dome ($893 \pm 13$ ka, Germa et al., 2011) shows a transmissivity value of 1.6 10-3 m$^2$ s-1, and confirms our opinion of a high hydraulic conductivity for theses domes – and higher that the transmissivity/hydraulic conductivity of the three studied geological formations. Furthermore, eruptive mechanisms of andesite and basalt lava flows on one hand, and intrusive domes on the other hand, are different. Intrusive domes are characterized by vertical fractures as shown on figure 9. Those fractures allows groundwater recharge to penetrate in depth, in agreement with water balance calculation. This mechanism of intrusion also explains that lava intrusion cross cut impermeable formations. Our correlation between age and hydraulic properties is then valid for the same kind of rocks (ie. Andesite and basalt lava flows in the context of subduction zone volcanic arc island). Consequently, the comparison of time evolution of lava flows and domes properties cannot be considered. This will be precise in the revised version.

Additional references:

Arnaud, L., Lanini, S.: Impact du changement climatique sur les ressources en eau de Martinique. Openfile BRGM Report RP-62676-FR, 2014.

Ecker, A.: Groundwater behavior in Tenerife, volcanic island (Canary-Islands, Spain). J. Hydrol. 28 (1), 73–86, 1976.

Ingebritsen, S.E., Sanford, W.E., Neuzil, C.E.: Groundwater in Geologic Processes. Cambridge University Press, 365 pp., ISBN 0-521-49608,1998 Second Edition., 2006.

Stollsteiner, P., Taïlamé, A.L.: Détermination des seuils de vigilance des niveaux d'eau souterraine en Martinique. Openfile BRGM Report RP-66058-FR, 2017.

Vittecoq, B., Lachassagne, P., Lanini, S., Ladouche, B., Marechal, J.C., and Petit, V. : Élaboration d'un système d'information sur les eaux souterraines de la Martinique : identification et caractérisations quantitatives. Rapport final. BRGM/RP-55099-FR, 2007.

---

## Editor Decision (ED1)

[revised manuscript text omitted]